# BTLA contributes to acute-on-chronic liver failure infection and mortality through CD4+ T-cell exhaustion

Xueping Yu [1,2,10] ✉, Feifei Yang[1,10], Zhongliang Shen[1,10], Yao Zhang[1,10], Jian Sun[1], Chao Qiu [1], Yijuan Zheng[2], Weidong Zhao[3], Songhua Yuan[4], Dawu Zeng[5], Shenyan Zhang[1], Jianfei Long[6], Mengqi Zhu[1], Xueyun Zhang[1], Jingwen Wu[1], Zhenxuan Ma[1], Haoxiang Zhu[1], Milong Su[2], Jianqing Xu[4], Bin Li [7], Richeng Mao [1] ✉, Zhijun Su [2] ✉ & Jiming Zhang [1,8,9] ✉

B- and T-lymphocyte attenuator (BTLA) levels are increased in patients with hepatitis B virus-related acute-on-chronic liver failure (HBV-ACLF). This condition is characterized by susceptibility to infection and T-cell immune exhaustion. However, whether BTLA can induce T-cell immune exhaustion and increase the risk of infection remains unclear. Here, we report that BTLA levels are significantly increased in the circulating and intrahepatic CD4+ T cells from patients with HBV-ACLF, and are positively correlated with disease severity, prognosis, and infection complications. BTLA levels were upregulated by the IL-6 and TNF signaling pathways. Antibody crosslinking of BTLA activated the PI3K-Akt pathway to inhibit the activation, proliferation, and cytokine production of CD4+ T cells while promoting their apoptosis. In contrast, BTLA knockdown promoted their activation and proliferation. BTLA[-/-] ACLF mice exhibited increased cytokine secretion, and reduced mortality and bacterial burden. The administration of a neutralizing anti-BTLA antibody reduced *Klebsiella pneumoniae* load and mortality in mice with ACLF. These data may help elucidate HBV-ACLF pathogenesis and aid in identifying novel drug targets.

Acute-on-chronic liver failure (ACLF) is characterized by the acute deterioration of liver function, with organ/system failure and a short- to medium-term mortality of 50–90%[1]. In China and other Asian countries, hepatitis B virus (HBV) infection remains the leading cause of ACLF, accounting for 80–85% of cases. Infection is one of the major causes of ACLF; moreover, patients with ACLF not triggered by infection are at a high risk of acquiring infections[2]. According to a study in China, 81.2% of patients with ACLF developed bacterial and fungal infections during their hospital stays, and 26.6% had a second infection[3]. The European Association for the Study of the Liver (EASL)-

[1]Department of Infectious Diseases, Shanghai Key Laboratory of Infectious Diseases and Biosafety Emergency Response, Shanghai Institute of Infectious Diseases and Biosecurity, National Medical Center for Infectious Diseases, Huashan Hospital, Fudan University, 200040 Shanghai, China. [2]Department of Infectious Diseases, First Hospital of Quanzhou Affiliated to Fujian Medical University, 362000 Quanzhou, China. [3]Department of Laboratory Medicine, Clinical Medicine College, Dali University, 671000 Dali, China. [4]Shanghai Public Health Clinical Center and Institutes of Biomedical Science, Shanghai Medical College, Fudan University, 200040 Shanghai, China. [5]Department of Hepatology, the First Affiliated Hospital, Fujian Medical University, 350000 Fuzhou, China. [6]Department of Pharmacy, Huashan Hospital, Fudan University, 200040 Shanghai, China. [7]Shanghai Institute of Immunology, Shanghai JiaoTong University School of Medicine, 200040 Shanghai, China. [8]Key Laboratory of Medical Molecular Virology (MOE/MOH), Shanghai Medical College, Fudan University, 200040 Shanghai, China. [9]Department of Infectious Diseases, Jing'An Branch of Huashan Hospital, Fudan University, 200040 Shanghai, China. [10]These authors contributed equally: Xueping Yu, Feifei Yang, Zhongliang Shen, Yao Zhang. ✉e-mail: xpyu15@fudan.edu.cn; njxiaomao@163.com; su2366@sina.com; jmzhang@fudan.edu.cn

Chronic Liver Failure (CLIF) reported that 66.1% of patients with ACLF presented with infections at diagnosis or during follow-up[4]. Frequent infection leads to accelerated deterioration of liver function and increases the mortality of ACLF[4]. However, the mechanism underlying susceptibility to infection in patients with ACLF has not been fully elucidated.

HBV-ACLF is often associated with a systemic inflammatory response (SIRS), which may be triggered by infection. Once a SIRS has manifested, the compensatory anti-inflammatory response (CARS) can be jump-started with many pro- and anti-inflammatory immune cells, molecules, and cytokines. The inappropriate CARS may cause the dysfunction and exhaustion of both innate and adaptive immune responses and decrease their ability to remove pathogens, a phenomenon known as immune exhaustion (or immune paralysis)[5]. Several studies have shown that immune exhaustion may be responsible for the increased risk of infection and is an independent predictor of death in patients with ACLF[6–8].

Previous studies have shown that an excess of regulatory cells (such as Treg and monocytic myeloid-derived suppressor cells) or coinhibitory molecules (such as programmed cell death protein-1 [PD-1], MER receptor tyrosine kinase, cytotoxic T-lymphocyte antigen 4 [CTLA-4], and T-cell immunoglobulin and mucin-domain containing-3 [TIM-3]) contribute to immune exhaustion in the progress of CARS[6,7,9,10]. However, the components involved in CARS are complex, and blocking PD-1 or CTLA-4 alone cannot completely restore the function of T cells. Therefore, it is necessary to further examine the role of other cells and molecules in immune exhaustion.

Recently, another molecule with negative immunosuppressive function has garnered attention: B- and T-lymphocyte attenuator (BTLA), a member of the CD28 immunoglobulin superfamily that is structurally related to CTLA-4 and PD-1. BTLA inhibits T/B cell activation and the secretion of cytokines[11] and plays an important role in autoimmune diseases, organ transplantation, and cancer[12–14]. In addition, BTLA-deficient mice have exhibited enhanced pathogen clearance compared to wild-type (WT) mice in the early phase of infection, while agonistic anti-BTLA antibodies have been shown to rescue mice from lipopolysaccharide (LPS)-induced endotoxic shock. These findings suggest that BTLA also plays a crucial role in the immune response against infectious pathogens[15,16].

Yang et al. demonstrated that upregulated BTLA expression induced by mouse hepatitis virus 3 (MHV-3) enhances the vitality and function of macrophages and causes acute liver failure (ALF), whereas blocking the BTLA signaling pathway can weaken the function of macrophages and reduce the fatality rate[17]. Xu et al. also showed that the expression of herpes virus entry mediator (HVEM)/BTLA was significantly increased in the liver tissue of patients with HBV-ACLF, but its effect and relevance were not clarified[18]. Further, MHV-3-induced liver disease may not represent the real HBV-ACLF. Moreover, little is known regarding the mechanisms underlying BTLA expansion in patients with HBV-ACLF, and whether BTLA-deficient or agonistic anti-BTLA antibodies can rescue T-cell immune exhaustion and reduce the risk of infection remains to be investigated.

Here, we aim to analyze BTLA expression in patients with HBV-ACLF and evaluate its relationship with disease severity, prognosis, and infection complications. Furthermore, we determine whether activating the BTLA signaling pathway in vivo or in vitro can account for T-cell exhaustion. Our findings identify an immunotherapeutic target to reduce susceptibility to infections and mortality in patients with HBV-ACLF.

## Results
### Expression of BTLA on CD4+ T cells and HVEM on dendritic cells (DC) is elevated synchronously in HBV-ACLF
To determine whether BTLA is involved in the pathogenesis of HBV, we measured BTLA expression in peripheral blood mononuclear cells

(PBMC) from 175 HBV-infected patients and 90 normal controls (NC). We found that the mean fluorescence intensity (MFI) and frequency of BTLA expression in peripheral blood CD4+ T cells (but not on CD8+ T cells) were significantly upregulated in patients with HBV-ACLF compared with NC and patients with chronic hepatitis B (CHB) (Fig. 1a, b). However, the frequency (but not the MFI) of BTLA expression on CD4+ T cells was slightly increased in patients with CHB compared with that in NC. When the inclusion criteria of ACLF agreed with the North American Consortium for the Study of End-stage Liver Disease (NACSELD)[19] and the EASL-CLIF[20] standards, the results also showed that BTLA was highly expressed on CD4+ T cells (but not on CD8+ T cells) from patients with HBV-ACLF (Supplementary Fig. 1a, b). Except for patients with HBV-ACLF, those with alcohol-induced ACLF and cirrhosis showed high expression of BTLA on CD4+ T cells (but not on CD8+ T cells), while patients with primary biliary cholangitis did not exhibit high expression of BTLA (n = 4; Supplementary Fig. 2a, b). Additionally, both CHB and HBV-ACLF patients displayed increased BTLA expression on natural killer (NK) cells but not on DC (Supplementary Fig. 2d, f). Notably, the elevated BTLA expression in patients with HBV-ACLF was predominant on the effector of memory T cells subtype and Th1, Th2, Th9, Th17, Th22, Th17-Th1, Tfh, and Treg subgroups of CD4+ T cells, but not on other CD4+ T subtypes (T effector memory RA positive cells, naive T cells, central memory T cells; Supplementary Fig. 3a–f).

As peripheral blood does not effectively reflect the true condition of the liver, we further investigated the distribution of BTLA expression on the liver-infiltrating lymphocytes (LIL) of patients with HBV-ACLF. Unexpectedly, the frequency of BTLA expression on LIL CD4+ T cells (but not on CD8+ T cells) was also significantly increased in HBV-ACLF patients than in CHB patients and NC (Fig. 1c; Supplementary Fig. 2c). Additionally, we determined BTLA expression protein in liver tissues by immunofluorescence and found that the frequency of double-positive CD4+BTLA+ cells was increased in HBV-ACLF patients compared with those in NC and CHB patients (Supplementary Fig. 4a).

Next, we determined whether the expression of HVEM, a BTLA-specific receptor, on T/NK cells and DC is correspondingly increased in patients with HBV-ACLF. We observed that HVEM expression on CD80/CD86+ DC increased in HBV-ACLF patients compared with that in NC and CHB patients (Fig. 1d), whereas HVEM expression on T cells decreased (Supplementary Fig. 2e, f).

### Upregulation of BTLA expression on CD4+ T cells was correlated with disease progression, prognosis, and infectious complications in patients with HBV-ACLF
To investigate whether BTLA expression was associated with the severity of the disease, we assessed the relationship between CD4+BTLA+ T cells from PBMC and clinical parameters. The MFI of BTLA expression on CD4+ T cells exhibited positive correlations with various disease severity indicators, including Child-Pugh scores[21], model for end-stage liver disease (MELD) scores[22], CLIF-Sequential Organ Failure Assessment (SOFA)[20], CLIF-C ACLFs[23], and Chinese Group on the Study of Severe Hepatitis B (COSSH)-ACLFs[24] (Fig. 1e; Supplementary Fig. 1c). Additionally, it was positively associated with physiological and biochemical indices of liver injury, such as total bilirubin (TBil) and international normalized ratio (INR) (Supplementary Fig. 4d), and systemic inflammation (neutrophil count, C-reactive protein (CRP), and procalcitonin (PCT)) (Supplementary Fig. 4g), but negatively correlated with compensatory indices of liver function (Albumin (ALB) and cholinesterase (CHE)) (Supplementary Fig. 4e). Moreover, during the progression of HBV-ACLF, the MFI of BTLA expression on CD4+ T cells increased in Grade 3 compared with that in Grade 2 and Grade 1 (Fig. 1f; Supplementary Fig. 1e). Next, we examined the distribution of CD4+BTLA+ T cells in the complications of patients with HBV-ACLF. We found that the MFI of BTLA expression on CD4+ T cells increased in co-infected patients compared with that in

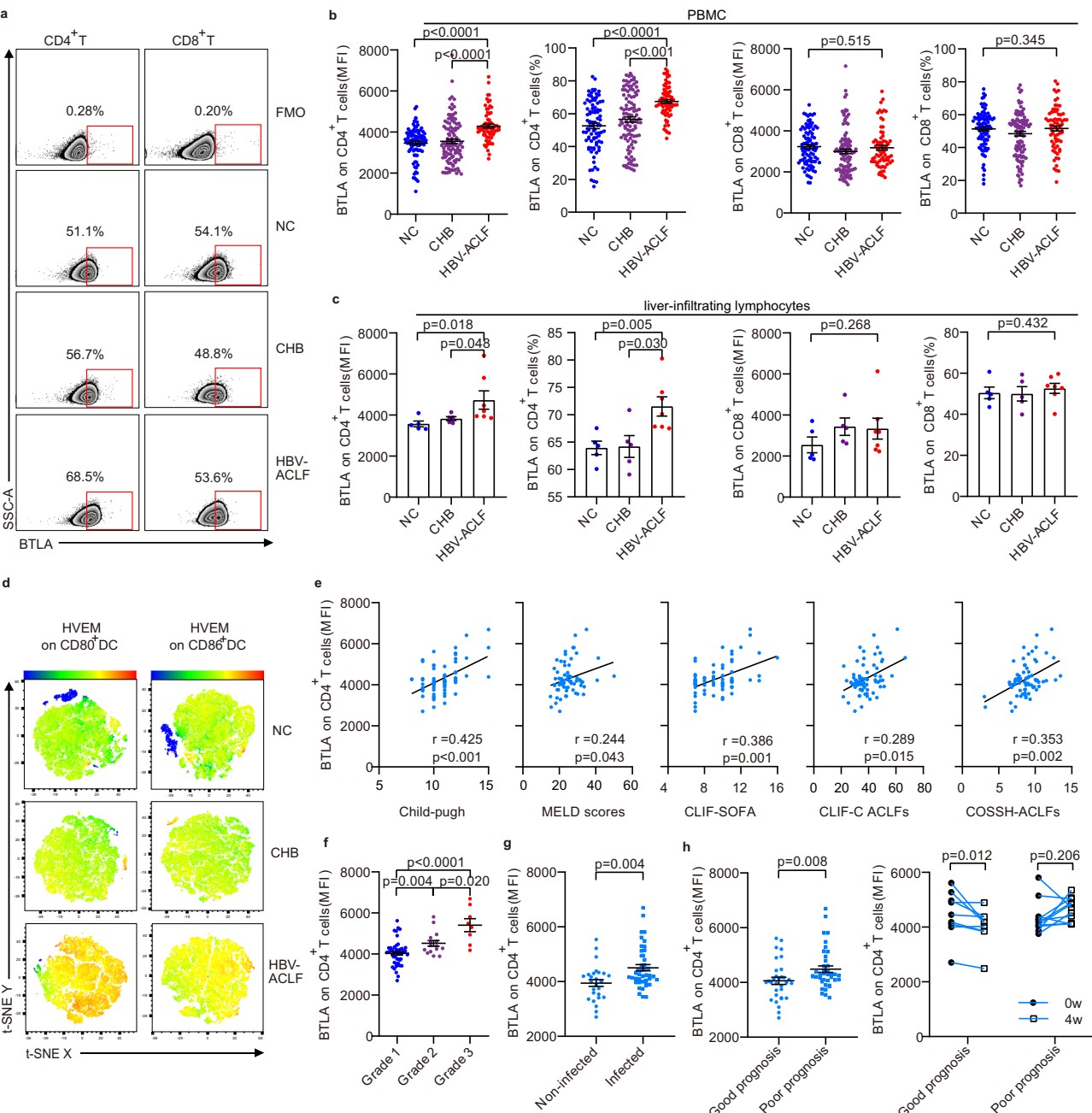

**Fig. 1 | BTLA expression was significantly increased on circulating and intrahepatic CD4+ T cells in patients with HBV-ACLF and was positively correlated with prognosis and infectious complications. a** Flow cytometry diagram of BTLA expression on peripheral blood CD4+ T and CD8+ T cells in NC, CHB, and HBV-ACLF patients. Patients with HBV-ACLF had significantly increased expression of BTLA on peripheral blood CD4+ T cells (**b**, NC: $n = 90$ donors, CHB: $n = 104$ donors, HBV-ACLF: $n = 71$ donors) and intrahepatic CD4+ T cells (**c**, NC: $n = 5$ donors, CHB: $n = 5$ donors, HBV-ACLF: $n = 7$ donors) compared to NC and CHB patients, while there was no significant difference in the BTLA expression on CD8+ T cells from PBMC or from liver-infiltrating lymphocytes. **d** T-distributed stochastic neighbor embedding (t-SNE) was used to illustrate the increased expression of HVEM on CD80/CD86+ dendritic cells in patients with HBV-ACLF. **e** The mean fluorescence intensity (MFI)

of BTLA expression on CD4+ T cells was positively correlated with the severity of the disease (Child-Pugh, MELD scores, CLIF-SOFA, CLIF-C ACLFs, and COSSH-ACLFs, $n = 71$ donors). **f** Changes in the MFI of CD4+BTLA+ T cells in the progression of HBV-ACLF ($n = 71$ donors). The MFI of CD4+BTLA+ T cells was increased in the coinfection patients (**g**, $n = 44$ donors) and poor prognosis group (**h**, left, $n = 39$ donors) compared with that in non-infected patients ($n = 27$ donors) and in those with good prognosis ($n = 32$ donors), respectively. **h** right HBV-ACLF patients with good prognosis ($n = 9$ donors), but not those with poor prognosis ($n = 11$ donors), had a significantly decreased MFI of CD4+BTLA+ T cells after 4 weeks of treatment. Data were calculated as mean ± SEM (**b, c, f, g, h**), Kruskal-Wallis H test followed by Dunn's multiple comparison test (**b, c, f**), Mann–Whitney U test (**g, h** left) and Wilcoxon test (**h**, right). A two-sided $P < 0.05$ was considered significant.

non-infected patients (Fig. 1g), while no significant differences were noted in patients with HBV-ACLF with or without ascites complications, as well as in Hepatitis B virus e antigen (HBeAg)-positive or HBeAg-negative patients (Supplementary Figs. 1d, 4b).

To investigate whether BTLA expression is associated with ACLF prognosis, patients were first divided into two groups based on

whether they experienced a good or poor prognosis. The good prognosis group included patients whose liver function and blood coagulation gradually recovered and who were still alive after 6 months. The poor prognosis group included patients whose liver function deteriorated progressively and who died or received liver transplantation with poor response to comprehensive treatment within 6 months. The

details of the two groups have been described in our previous study[25]. We found that the MFI of BTLA expression on CD4[+] T cells was increased in the poor prognosis group compared with that in the good prognosis group (Fig. 1h; Supplementary Fig. 1d). However, following comprehensive treatment, it gradually decreased (Supplementary Figs. 1f and 4c). Importantly, the HBV-ACLF patients in the good prognosis group had significantly decreased CD4[+]BTLA[+] T cells at 4 weeks post-treatment, while the poor prognosis group exhibited no significant change (Fig. 1h).

## Phenotypical characteristics of BTLA[+]CD4[+] T cells

To characterize the phenotype of BTLA[+]CD4[+] T cells, we compared the expression profiles of BTLA[+]CD4[+] and BTLA[-]CD4[+] T-cell subsets in peripheral blood from patients with HBV-ACLF sorted by flow cytometry and then subjected to trace RNA sequencing (RNA-Seq; Fig. 2a). As shown in Fig. 2b, BTLA[+]CD4[+] T cells expressed higher levels of CD127, HVEM, CD28, CD7, inducible co-stimulatory molecule (ICOS), TIM-3, CD40, CD25, PD-1, CD64, lymphocyte-activation gene 3 (LAG-3), and C-C chemokine receptor 5 (CCR5), but lower levels of CD57 and CTLA-4 than BTLA[-]CD4[+] T cells. These results were confirmed by flow cytometry in the peripheral blood of NC, CHB, and HBV-ACLF patients, whose expression of CD7 (T-cell interactions), CD127 (T-cell survival), and CD25 (activation) was significantly increased on BTLA[+]CD4[+] T cells compared with that in BTLA[-]CD4[+] T cells (Fig. 2c).

To determine the differentiation phenotype of BTLA[+] CD4[+] T cells, we examined the frequencies of the subtypes of effector memory cells re-expressing CD45RA (TEM-RA), Tnaive, central memory (Tcm), and effector memory (Tem) of BTLA[+]CD4[+] T cells and BTLA[-]CD4[+] T cells in peripheral blood of NC, CHB, and HBV-ACLF patients. In NC, the proportions of Tem and TEM-RA subtypes (representing differentiated subsets of CD4[+] T cells) on BTLA[+]CD4[+] T cells were higher than on BTLA[-]CD4[+]T cells, while there were no significant differences in the proportion of Tnaive and Tcm subtypes. Moreover, there were no significant differences in the proportion of CD4[+] T subtypes between BTLA[+]CD4[+] T cells and BTLA[-]CD4[+] T cells in CHB and HBV-ACLF patients. (Fig. 2d, e). We also found that compared with BTLA[-]CD4[+]T cells, BTLA[+]CD4[+]T cells secreted higher levels of IFN-γ, IL-2, and TNF-α, and had higher proliferation ability (Fig. 2f, g).

## Elevated BTLA expression on CD4[+] T cells was induced by circulating inflammation cytokines in HBV-ACLF

Based on the inflammatory microenvironment that regulates BTLA expression[12], we investigated the role of circulating cytokines. Except for a few cytokines that were not elevated in HBV-ACLF, the levels of pro-inflammatory cytokines (interleukin [IL]-1β, IL-6, IL-22, and tumor necrosis factor [TNF]-α) and anti-inflammatory cytokines (IL-10 and IL-37) were markedly increased in patients with HBV-ACLF compared with NC and CHB patients (Fig. 3a). This suggests the onset of CARS in response to excessive SIRS, which is consistent with BTLA expression in patients with HBV-ACLF. Furthermore, we found that only the levels of IL-6 and TNF-α were significantly and positively correlated with BTLA expression on CD4[+] T cells (Supplementary Fig. 5a). Therefore, we hypothesized that these elevated pro-/anti- inflammatory cytokines might induce BTLA expression in patients with HBV-ACLF. We used recombinant human (rh) IL-1β, rhIL-6, rhIL-22, rhIL-10, rhIL-37, and rhTNF-α to stimulate the PBMC from NC in vitro. We observed that BTLA expression on CD4[+] T cells was higher on day 3 than on day 5 or day 7 (Supplementary Fig. 5b). Only rhIL-6 and rhTNF-α could significantly increase the expression of BTLA on CD4[+] T cells in a dose-dependent manner, but not that of other recombinant cytokines (Fig. 3b). We also demonstrated that rhIL-6 and rhTNF-α could induce the upregulation of *BTLA* messenger (mRNA) levels in a dose-dependent manner (Supplementary Fig. 5c). More importantly, we found that the decreased MFI of BTLA expression on CD4[+] T cells was accompanied by decreased levels of IL-6 and TNF-α induced by

comprehensive treatment (such as antiviral, anti-inflammatory, and liver-protecting treatment) in nine patients with HBV-ACLF (Fig. 3c). To provide a mechanistic explanation for the upregulation of BTLA expression in patients with HBV-ACLF, we further investigated the effect of HBV-ACLF plasma on BTLA expression in CD4[+] T cells purified from NC. As shown in Fig. 3d, HBV-ACLF plasma exposure treatment resulted in a significant increase in BTLA expression on CD4[+] T cells compared to NC plasma treatment, which was reversed by anti-IL-6 and/or anti-TNF-α (unreactive to anti-IL-1β).

IL-6 and TNF-α are known to play their biological roles through STAT3 and NF-κb signaling, respectively. We found that the levels of *Stat3* mRNA in patients with HBV-ACLF were significantly higher than those in NC and patients with CHB, and rhIL-6 significantly increased the levels of *Stat3* mRNA dose-dependently (Supplementary Fig. 5d). The stimulating effect of rhIL-6 or rhTNF-α on inducing BTLA expression on CD4[+] T cells could be abolished or partially weakened by anti-stat3 or anti-NF-κb inhibitors, respectively (Supplementary Fig. 5e).

## Elevated BTLA induces CD4[+]T-cell exhaustion by inhibiting activation, proliferation, and secretory cytokines or by promoting apoptosis of CD4[+] T cells

T-cell immunodeficiency is known to be present in patients with HBV-ACLF. Our results also revealed that HBV-ACLF patients displayed decreased activation, proliferation, and secretion of cytokines but had an increased apoptosis rate in CD4[+] T cells, suggesting that the function of CD4[+] T cells tends to be exhausted (Supplementary Fig. 6a–d). Moreover, HBV-ACLF patients with poor prognosis displayed lower plasma levels of Th1-like [interferon (IFN)-γ and TNF-α], Th2-like cytokines (IL-12 and IL-4), and other chemokines (granulocyte-macrophage colony stimulating factor, macrophage derived chemokine, and macrophage inflammatory protein-1α), but higher levels of IL-10 compared with HBV-ACLF patients with good prognosis (Supplementary Fig. 6e). BTLA has been shown to maintain T-cell tolerance by inhibiting T-cell proliferation and the production of IFN-γ and IL-10[26]. However, the role of BTLA on CD4[+] T-cell exhaustion in HBV-ACLF patients has not been addressed clearly. We utilized CD25 and CD69 as well as CD38 to define activated T cells and used an agonistic anti-BTLA monoclonal antibody (Ab) to crosslink BTLA. Crosslinking of BTLA using BTLA Ab for 1 day resulted in the strongest suppression of CD4[+] T-cell activation (Supplementary Fig. 7a). However, crosslinking of BTLA did not result in obvious changes in the expression of other immune checkpoints on CD4[+] T cells, such as PD-1, CTLA-4, TIM-3, and T-cell immunoglobulin and immunoreceptor tyrosine-based inhibitory motif domain (TIGIT; Supplementary Fig. 7b).

We also isolated BTLA[+]CD4[+] T cells from the peripheral blood of NC and found that crosslinked BTLA markedly inhibited the expression of activation markers (CD25, CD38, and CD69), the production of IFN-γ, IL-2, and TNF-α induced by phorbol 12-myristate 13-acetate/ionomycin, and the proliferation of BTLA[+]CD4[+] T cells, while promoting the apoptosis of BTLA[+]CD4[+] T cells (Fig. 4a–d). A similar phenomenon was observed in the CD4[+] T cells from the peripheral blood of NC, CHB, and HBV-ACLF patients (Supplementary Fig. 8a–h).

## Short hairpin RNA (shRNA) knockdown BTLA increases CD4[+] T-cell activation, production of cytokines, and proliferation, and abrogates the apoptosis of CD4[+] T cells

To identify whether reduced expression of BTLA in CD4[+] T cells is beneficial to restoring their function, we constructed three specific shRNAs targeting BTLA, all of which could inhibit BTLA expression on CD4[+] T cells from the PBMC of NC (Supplementary Fig. 7c). Although BTLA shRNA reduced BTLA expression on CD4[+] T cells in a dose-dependent manner, it was able to display a potent inhibitory effect at a concentration of 200 nM (Fig. 5a). After crosslinking BTLA by an agonistic anti-BTLA monoclonal Ab, the proliferation of CD4[+] T cells was

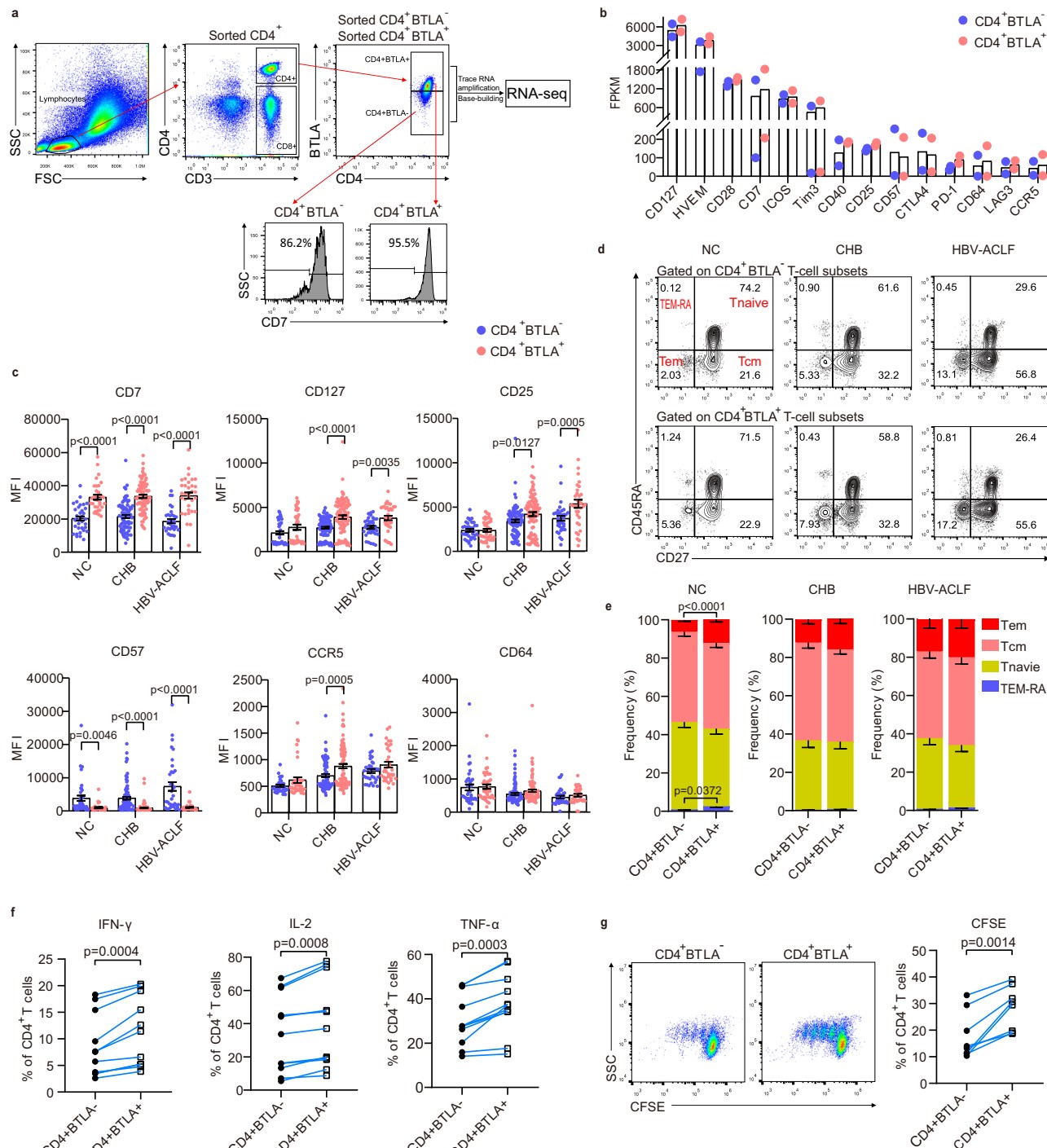

**Fig. 2 | Phenotypic characteristics of BTLA⁺CD4⁺ T cells. a** Flow cytometry was used to sort different BTLA subpopulations, followed by RNA-seq. **b** Differently expressed genes (DEGs) between BTLA⁺CD4⁺ T-cell subsets and BTLA⁻CD4⁺ T-cell subsets (all *n* = 2 donors). **c** Validation of DEGs in BTLA⁺CD4⁺ T-cell subsets and BTLA⁻CD4⁺ T-cell subsets in NC (*n* = 38 donors), CHB (*n* = 94 donors) and HBV-ACLF patients (*n* = 35 donors). **d** Representative flow cytometry of BTLA⁺CD4⁺ T-cell subsets and BTLA⁻CD4⁺ T-cell subsets. **e** Differentiation phenotypes of BTLA⁺ CD4⁺

T cells in NC (*n* = 26 donors), CHB (*n* = 17 donors), and HBV-ACLF patients (*n* = 13 donors). Compared with BTLA⁻CD4⁺ T cells, BTLA⁺CD4⁺T cells secreted higher levels of IFN-γ, IL-2, and TNF-α (**f**, *n* = 10 donors), and showed a greater proliferation ability (**g**, *n* = 10 donors). Data were calculated as mean ± SEM (**c**), Two-way ANOVA followed by Sidak's multiple-comparison test (**c**), Mann–Whitney test (**e**), and Wilcoxon test (**f, g**). A two-sided *P* < 0.05 was considered significant.

significantly increased in the BTLA shRNA group than in the control shRNA group (Fig. 5b; Supplementary Fig. 7d). Moreover, BTLA shRNA increased the production of IFN-γ, IL-2, and TNF-α and the expression of CD25, CD38, and CD69 in CD4⁺ T cells (Fig. 5d, e). Conversely, BTLA shRNA-treated CD4⁺ T cells tended to exhibit lower apoptosis levels than control shRNA-treated CD4⁺ T cells (Fig. 5c).

## BTLA may activate the phosphoinositide-3-kinase/Akt/glycogen synthase kinase-3 (PI3K-Akt-GSK-3β) pathway to induce CD4⁺T-cell exhaustion

To explore the mechanisms by which BTLA induces CD4⁺ T-cell exhaustion in HBV-ACLF patients and to seek drug targets for recovering the function of CD4⁺ T cells, we analyzed gene expression

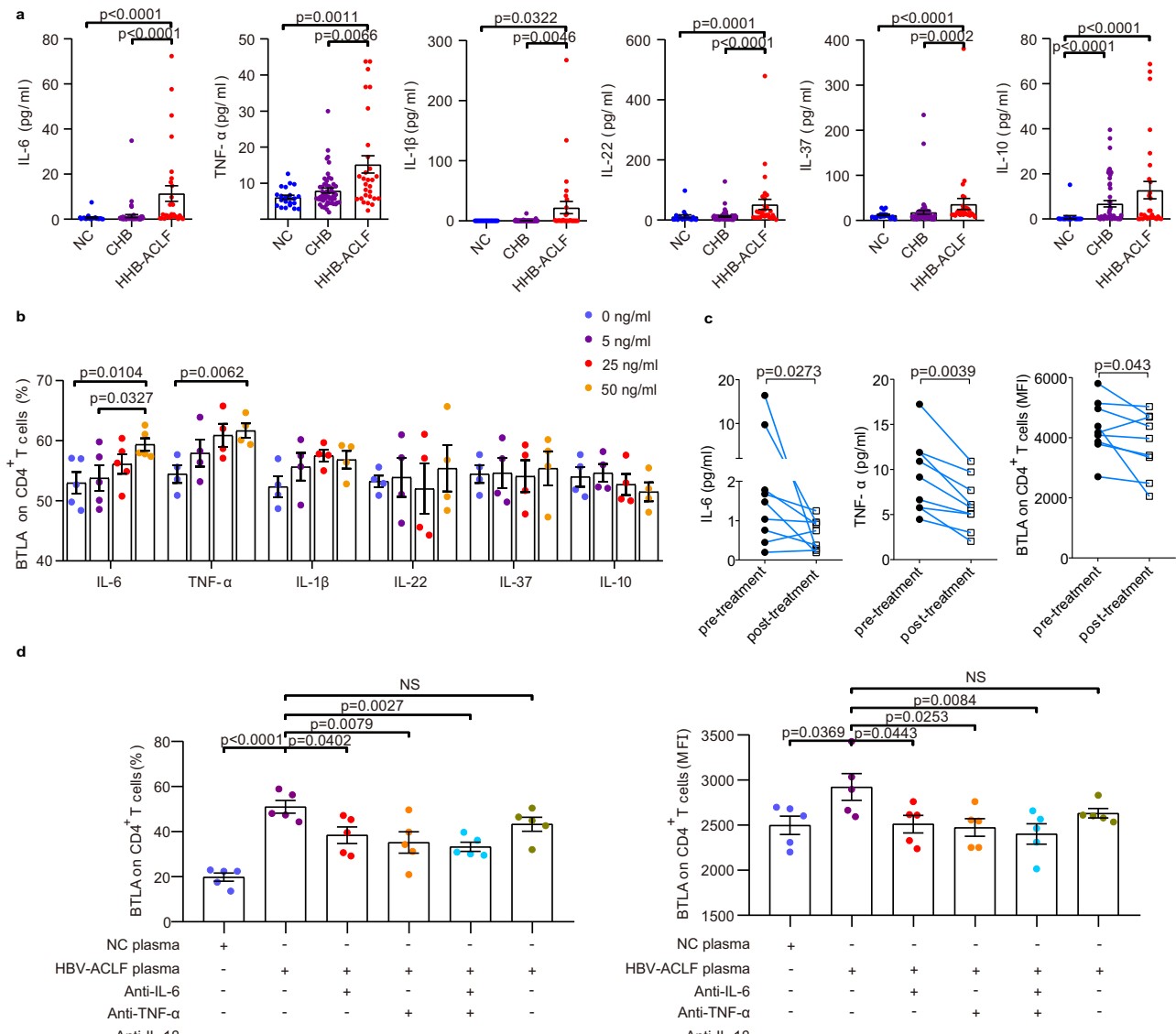

**Fig. 3 | Elevated BTLA expression on CD4+ T cells was induced by circulating inflammation cytokines in HBV-ACLF. a** The levels of pro-inflammatory cytokines (IL-1β, IL-6, IL-22, and TNF-α) and anti-inflammatory cytokines (IL-10 and IL-37) were markedly increased in HBV-ACLF patients ($n = 29$ donors) compared with those in NC ($n = 21$ donors) and CHB patients ($n = 53$ donors). **b** Only rhIL-6 and rhTNF-α significantly increased BTLA expression on CD4+ T cells in a dose-dependent manner ($n = 4$ donors). **c** Comprehensive treatment ($n = 9$ donors) decreased the levels of IL-6 and TNF-α, as well as BTLA expression on CD4+ T cells. **d** Exposure of

purified CD4+ T cells from NC to HBV-ACLF plasma ($n = 13$ donors) resulted in a higher BTLA expression on CD4+ T cells compared with exposure of PBMC to NC plasma, while HBV-ACLF plasma combined with anti-IL-6 and/or anti-TNF-α resulted in decreased BTLA expression on CD4+ T cells. Data were calculated as mean ± SEM (**a, b, d**), Kruskal-Wallis H test followed by Dunn's multiple comparison test (**a**), Wilcoxon test (**c**), and one-way ANOVA followed by Tukey's (**b**) or Dunnett's multiple comparison test (**d**). A two-sided $P < 0.05$ was considered significant.

profiles in PBMC from NC, CHB, and HBV-ACLF patients with or without BTLA Ab treatment through next-generation high-throughput RNA-Seq. Anti-BTLA crosslinking resulted in a larger number of differentially expressed genes (DEGs; Supplementary Fig. 9a–i), of which 237 were shared by the above three groups (Fig. 6a). Kyoto Encyclopedia of Genes and Genomes (KEGG) analysis was performed according to these 237 DEGs, and six signaling pathways, namely PI3K-Akt, RAS, cGMP-PKG, toll-like receptor, nucleotide-binding oligomerization domain-like receptor, and phospholipase D, were upregulated or downregulated. Notably, the PI3K-Akt pathway showed the most significant difference as it exhibited the largest number of upregulated DEGs (Fig. 6b, c). Subsequently, through western blotting, it was determined that anti-BTLA crosslinking could activate the PI3K-Akt-GSK-3β pathway in purified CD4+ T cells (Fig. 6d). As previously reported[11,14], anti-BTLA crosslinking may also cause the

phosphorylation of the Src-homology domain 2 (SH2)-containing proteins tyrosine phosphatase-1 (SHP-1) and SHP-2 to activate suppression signaling (Fig. 6d).

## BTLA contributes to CD4+ T-cell immune exhaustion and increases mortality and infection rates in the ACLF model induced by Concanavalin A (ConA)

Zhang et al. successfully developed an animal model of ACLF immune status from SIRS to CARS through administration of ConA (15 mg/kg) five times every 48 h[27]. To prevent premature mice death due to acute liver injury, the dose and frequency of Con A injection were set as 8 mg/kg and eight times every 48 h, respectively (Fig. 7a). The results of a liver pathological analysis using hematoxylin-eosin staining showed that the ACLF was successfully established on days 8 and 14 in both WT and BTLA knockout (BTLA$^{-/-}$) mice (Fig. 7b; Supplementary

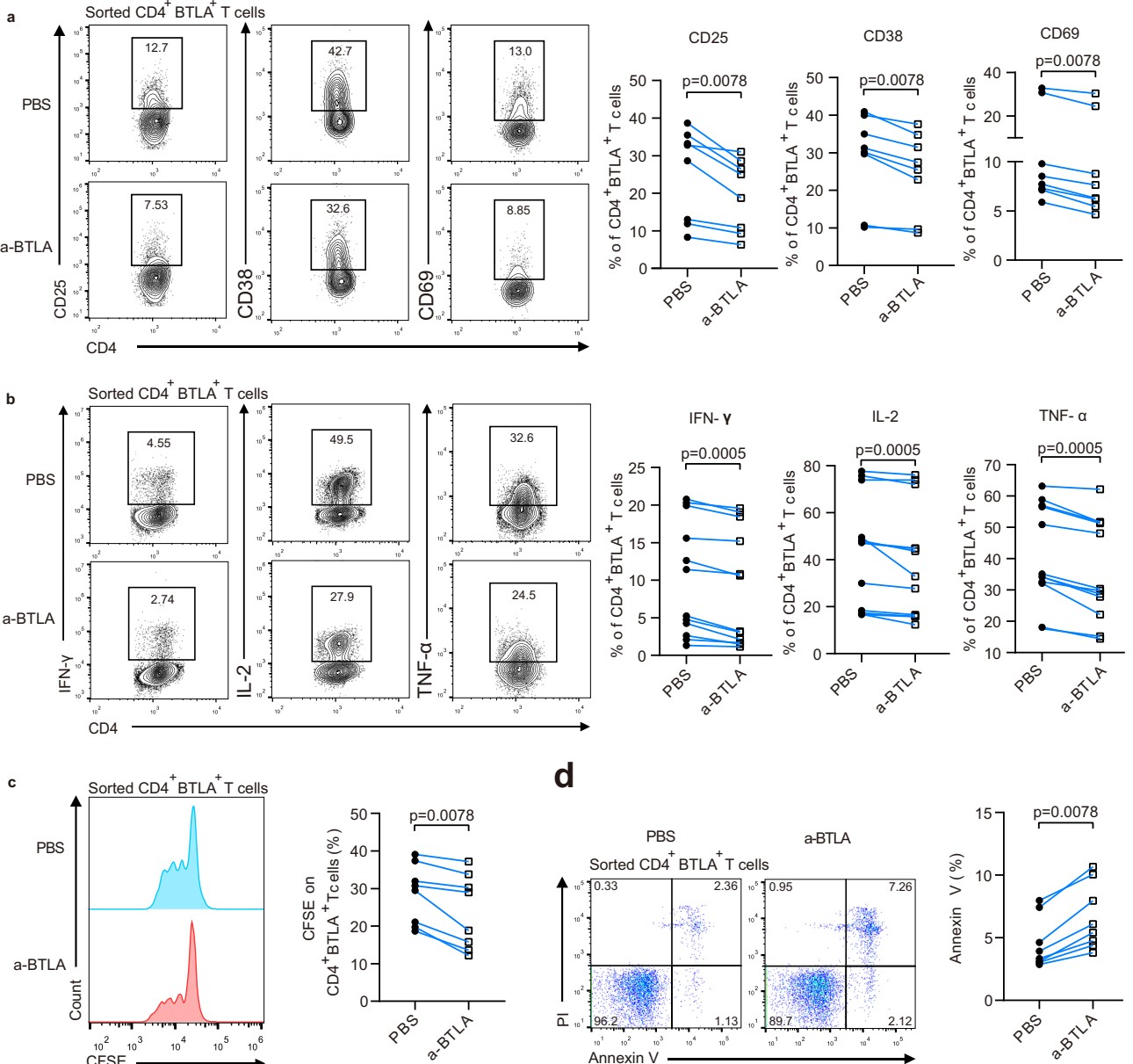

**Fig. 4 | Crosslinked BTLA markedly inhibited the activation, production of cytokines, and proliferation of BTLA⁺CD4⁺ T cells but promoted apoptosis.** **a** Flow cytometry for CD25, CD38, and CD69 (n = 8 donors), (**b**) production of IFN-γ, IL-2, and TNF-α induced by phorbol 12-myristate 13-acetate /ionomycin (n = 12

donors), (**c**, **d**) proliferation (n = 8 donors), and apoptosis of BTLA⁺CD4⁺ T cells from NC (n = 8 donors) in the control and crosslinked anti-BTLA groups. Wilcoxon test (**a**–**d**). A two-sided P < 0.05 was considered significant.

Fig. 10a–d). Unexpectedly, BTLA expression on CD4⁺/CD8⁺ T cells of peripheral blood from WT mice was significantly increased on days 8 and 14 (Fig. 7c; Supplementary Fig. 11a, b). Importantly, compared to baseline, the percentages and MFI of the activation molecules (CD25, CD38, and CD69) on CD4⁺ T cells was increased at days 2, 8, and 14, in both WT and BTLA⁻/⁻ mice following ConA injection; however, their expressions were higher in BTLA⁻/⁻ mice than in WT mice at day 14 (Fig. 7d; Supplementary Fig. 11c). Although the serum levels of cytokines (IL-6, IFN-γ and TNF-α) were slightly increased in BTLA⁻/⁻ mice than in WT mice (Supplementary Fig. 10e), the percentages and MFI of cytokines (IFN-γ and TNF-α) in CD4⁺ T cells were higher in BTLA⁻/⁻ mice than in WT mice at day 14 (Fig. 7e; Supplementary Fig. 11d). These results suggest that continuous ConA injection (eight times in 14 days) is more likely to induce immune exhaustion in WT ACLF mice than in BTLA⁻/⁻ ACLF mice.

To determine whether BTLA knockout improves the prognosis of ACLF mice or reduces the bacterial burden, we further constructed a sepsis model via cecal ligation and puncture (CLP) based on ConA-induced ACLF model (Fig. 7f). Following CLP, the mortality rate and bacterial DNA levels in whole blood were significantly decreased in BTLA⁻/⁻ mice compared with those in WT mice (Fig. 7g, h).

### BTLA contributes to CD4⁺ T-cell immune exhaustion and increased mortality and infection rates in the ACLF model induced by carbon tetrachloride (CCl₄)

Previous studies have shown that CCl₄ combined with LPS or *Klebsiella pneumoniae* (K.P.) injection can induce a mouse model of ACLF[15,28,29]. To better mimic ACLF conditions, we developed a mouse model with severe liver injury by combining chronic injury (CCl₄ injection), acute hepatic insult (injection of a double dose of CCl₄), and K.P. with or

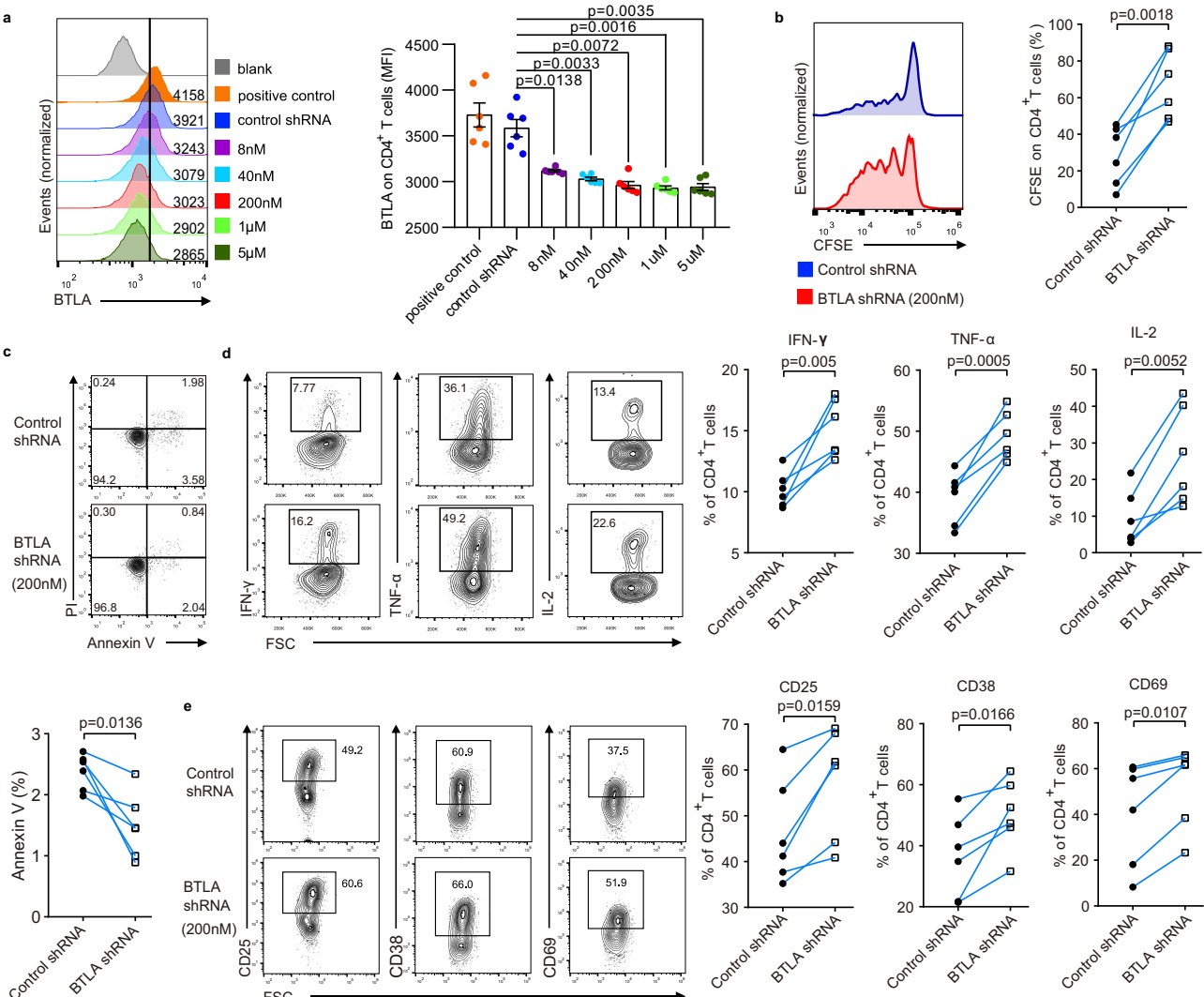

**Fig. 5 | BTLA knockdown increased activation, production of cytokines, and proliferation of CD4+ T cells but abrogated apoptosis. a** BTLA expression (left) and MFI (right) in freshly isolated PBMC activated for 3 days in the presence of control shRNA and varying concentrations of BTLA shRNA ($n = 5$ donors). **b** Carboxyfluorescein succinimidyl ester (CFSE) indicates the proliferation of the CD4+ T cells that were activated by anti-CD3/CD28 stimulation for 5 days in the presence of control shRNA and BTLA shRNA (left). The quantification (right) is presented as a division index ($n = 6$ donors). **c** Contour plots (up) and line graphs (down) display frequencies of Annexin V+CD4+ T cells in the presence of control

shRNA and BTLA shRNA ($n = 6$ donors). **d** Contour plots (left) and line graphs (right) display frequencies of IFN-γ, IL-2, and TNF-α-expressing cells on CD4+ T cells induced by PMA/ionomycin for 6 h in the presence of control shRNA and BTLA shRNA ($n = 6$ donors). **e** Expression of CD25, CD38, and CD69 on CD4+ T cells activated by anti-CD3/CD28 for 1 day in the presence of control shRNA and BTLA shRNA. All PBMC were isolated from NC ($n = 6$ donors). Data were calculated as mean ± SEM (**a**), one-way ANOVA followed by Dunnett's multiple comparison test (**a**), and Wilcoxon test (**b**–**e**). A two-sided $P < 0.05$ was considered significant.

without infection. As illustrated (Fig. 8a, b and Supplementary Fig. 12a–d), after continuous injection of $CCl_4$ for 8 weeks, a large number of inflammation/necrosis and fibrosis formation were observed in the liver of WT or BTLA$^{-/-}$ mice. Especially after injection of a double dose of $CCl_4$, inflammation and fibrosis increased, and alanine transaminase, aspartate aminotransferase, and total bilirubin were significantly enhanced, suggesting successful induction of the ACLF model. In both WT and BTLA$^{-/-}$ mice, the production levels of TNF-α, IL-2, and IFN-γ of CD4+ T cells decreased after 4 weeks of continuous $CCl_4$ injection and then began to gradually increase. Importantly, at weeks 6 and 8, BTLA$^{-/-}$ mice had higher expression levels of the above cytokines than WT mice (Fig. 8c). Similarly, after a double dose of $CCl_4$ injection, we also detected higher cytokine levels (TNF-α, IL-2, and IFN-γ) and slightly lower levels of IL-10 in the plasma of BTLA$^{-/-}$ mice compared with that of WT mice (Fig. 8d; Supplementary Fig. 12e). These results suggest that BTLA deficiency is not prone to immune exhaustion. After intraperitoneal injection of K.P., we observed that the survival rate of

BTLA$^{-/-}$ mice was higher than that of WT mice, and the survival rate of WT mice injected with anti-BTLA antibodies was also higher than that of WT mice injected with anti-IgG antibodies (Fig. 8e–h). As expected, the K.P. load in peripheral blood of both BTLA$^{-/-}$ mice and WT mice injected with anti-BTLA antibodies was significantly lower than that in the WT and anti-IgG injection groups (Fig. 8e–h).

## Discussion
HBV-ACLF is characterized by immune exhaustion and susceptibility to infectious diseases; however, the underlying mechanisms remain elusive. Our results indicate that 61.39% of patients with HBV-ACLF developed bacterial infections during their hospital stay. The elevated BTLA expression on CD4+ T cells, particularly the Tem subtype and all subgroups of CD4+ T cells, is a novel finding in the pathogenesis of immune exhaustion in HBV-ACLF. Importantly, the magnitude of BTLA expression on CD4+ T cells was positively correlated with the severity of disease, prognosis, and impaired antimicrobial response. BTLA

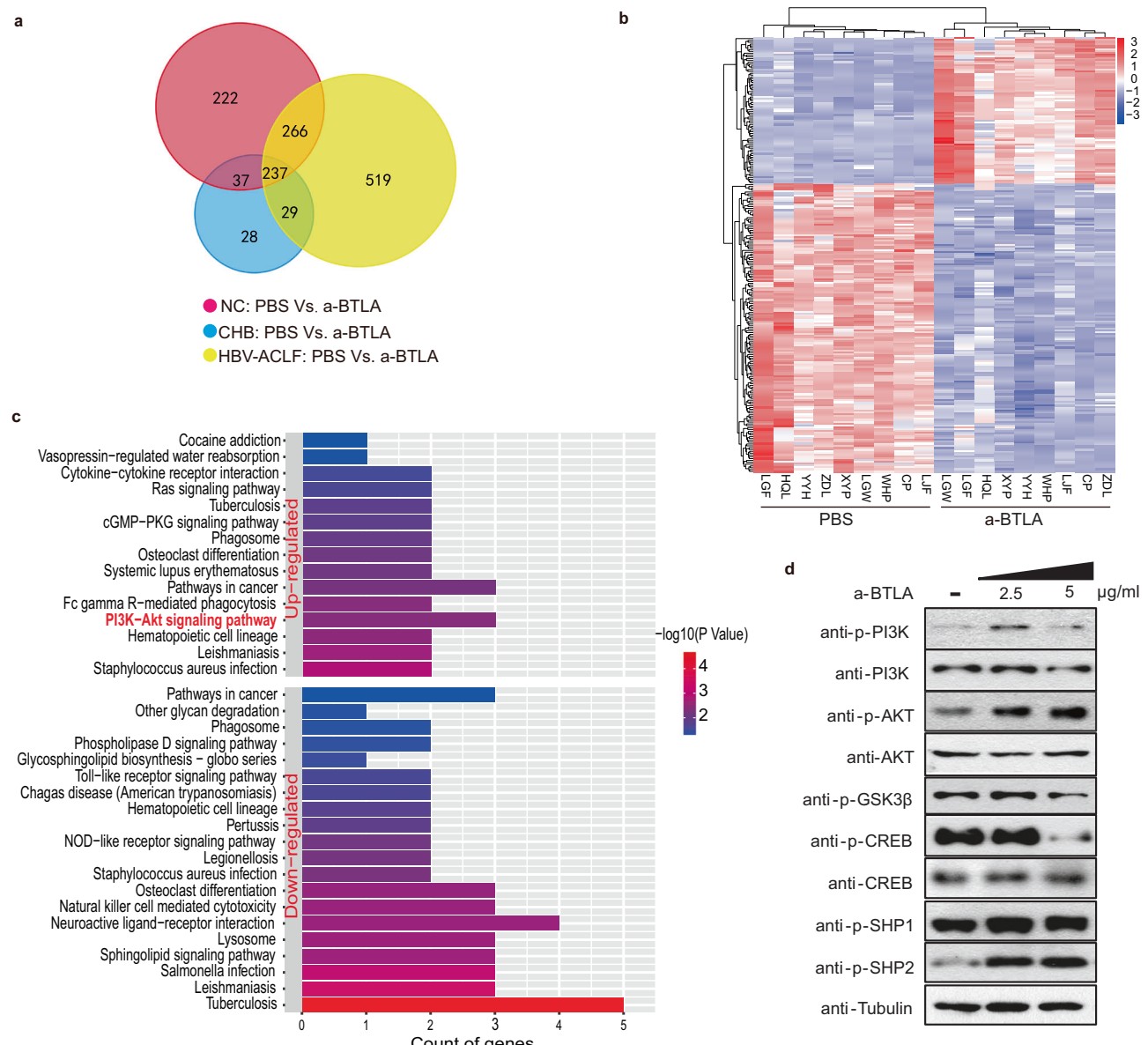

**Fig. 6 | Crosslinking of BTLA phosphorylated the SHP1/2 protein and activated the PI3K-Akt-GSK-3β pathway. a, b** Venn diagram (left) and heatmap (right) of DEGs, 237 of which were shared by NC ($n = 3$ donors), CHB ($n = 3$ donors) and HBV-ACLF patients ($n = 3$ donors). **c** KEGG pathway analysis showed that the PI3K-Akt pathway was significantly different, with the largest DEG in the upregulated signaling pathways. **d** Crosslinking of BTLA phosphorylates the SHP1/2 protein and activates the PI3K-Akt-GSK-3β pathway in purified CD4⁺ T cells. Data represent two experiments.

knockout or administration of a neutralizing anti-BTLA antibody rescued ACLF mice from the immune exhaustion state of CD4⁺ T cells and produced additional functional cytokines. These results suggest that BTLA may serve as a marker of T-cell exhaustion in HBV-ACLF.

It has been previously reported that the HBV-specific CD8⁺ T-cell immune response is very weak and below the detection level in patients with CHB[30], which may be related to the upregulation of multiple coinhibitory signaling pathways. Thimme et al. comprehensively analyzed the hierarchy of expression of nine coinhibitory receptors in HBV-specific CD8⁺ T cells and found that the hierarchy was dominated by PD-1, followed by KLRG1, 2B4, CD160, CTLA-4, TIM-3, BTLA, CD305, and LAG-3, suggesting that BTLA is required but not essential for weakening CD8⁺ T-cell immune response. Our demonstration of expression of BTLA on CD4⁺/CD8⁺ T cells was similar in NC and CHB patients, which is in agreement with other studies[31,32].

HBV-ACLF is a distinct condition that develops in patients with CHB. In 2012, Chen et al.[18] demonstrated that HVEM-BTLA signaling

significantly increased in the liver tissue of patients with HBV-ACLF; however, its role and clinical significance were not entirely elucidated. Our results showed that patients with HBV-ACLF exhibited upregulated BTLA expression on CD4⁺ T cells but not on CD8⁺ T cells in circulation PBMC and liver tissue compared with NC and CHB patients. Additionally, several studies have shown that BTLA and CTLA-4 are altered mainly on CD4⁺ T cells not on CD8⁺ T cells[7,33,34]. This is likely because: (1) BTLA expression is normally downregulated during human CD8⁺ T-cell differentiation to effector cytotoxic T cells; (2) CD4⁺ T cells are the main source of IL-10 and IL-21, which contribute to sustaining antiviral CD8⁺ T cells during chronic infection[35]; and (3) CD4⁺ T cells have more functional properties and differentiation potential than CD8⁺ T cells[35]. Therefore, under conditions of chronic HBV infection, characterized by many inflammatory cytokines (a cytokine storm) in HBV-ACLF, CD4⁺ T cells are more susceptible to immune dysfunction than CD8⁺ T cells and are more vulnerable to the effects of the inner inhibitory microenvironment (such as BTLA).

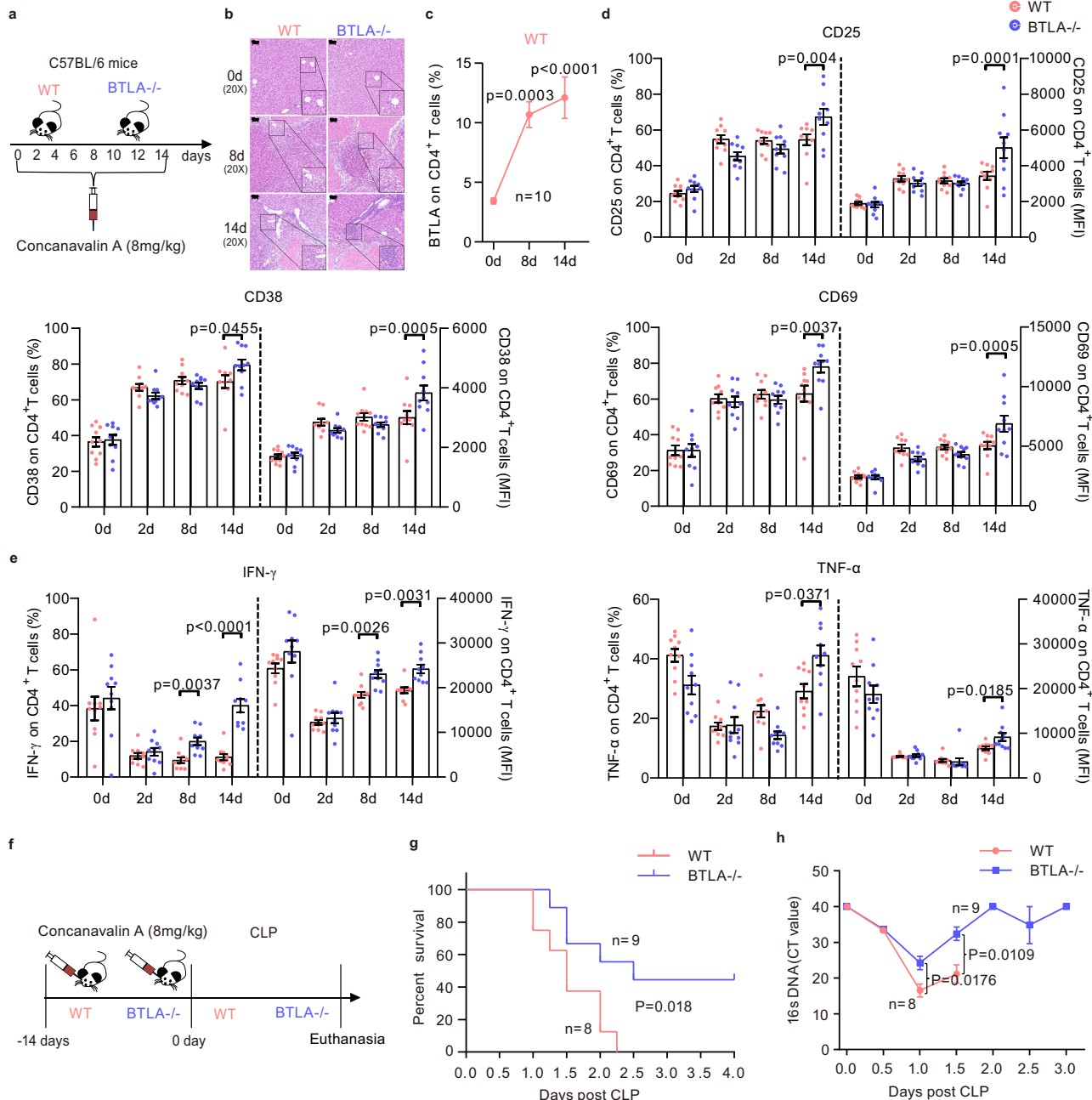

**Fig. 7 | BTLA expression contributed to immune exhaustion, increased mortality rate, and bacterial burden in the ACLF model induced by ConA.**
**a**, **b** Schematic diagram (left) of ACLF induced by continuous injection of ConA (8 mg/kg) and hematoxylin-eosin diagrams (right) of liver pathology at baseline and days 8 and 14 in WT ($n = 10$ mice) and BTLA[-/-] ($n = 10$ mice) C57BL/6 mice. **c** BTLA expression on CD4[+] T cells was significantly increased on days 8 and 14, compared to baseline, in WT mice ($n = 10$ mice) following ConA injection. **d** The percentages and MFI of activation indices (CD25, CD38, and CD69) were higher in BTLA[-/-] mice ($n = 10$ mice) than in WT mice ($n = 10$ mice) following ConA injection on day 14. **e** The levels of IFN-γ and TNF-α were higher in BTLA[-/-] mice ($n = 10$ mice) than in WT mice ($n = 10$ mice) at days 8 and (or) 14. **f** The cecal ligation and puncture (CLP) model of sepsis was established based on WT ($n = 8$ mice) and BTLA[-/-] ACLF mice ($n = 9$ mice) induced by continuous ConA injection. **g** Following CLP, BTLA[-/-] mice were significantly protected from mortality compared to WT mice. **h** Bacterial DNA levels in whole blood were significantly decreased in BTLA[-/-] ACLF mice ($n = 9$ mice) compared to those in WT mice ($n = 8$ mice) following CLP. Data were calculated as mean ± SEM (**c**–**e**), one-way ANOVA followed by Dunnett's multiple comparison test (**c**), Two-way ANOVA followed by Sidak's multiple-comparison test (**d**, **e**), Wilcoxon test (**h**), log-rank test for survival study (**g**). A two-sided $P < 0.05$ was considered significant.

Recent studies indicate that exhaustion of CD4[+] T cells caused by coinhibitory signals (such as PD-1, CTLA-4, and BTLA) and a lack of CD4[+] T cells contribute to CD8[+] T-cell immune dysfunction[36,37]. Elevated levels of BTLA on CD4[+] T cells will eventually result in CD8[+] T-cell immune disorders.

Inflammatory microenvironment components (such as IFN-γ and IL-7) may regulate BTLA expression[12,26]. Antoniades et al.[7] found that the expansion of CTLA-4, a coinhibitory molecule similar to BTLA, on CD4[+] T cells can be induced by the plasma of patients with ALF. Consequently, we hypothesize that the liver inflammatory microenvironment of HBV-ACLF may induce the compensatory expression of BTLA on CD4[+] T cells. In agreement with Antoniades et al., we found that the plasma of HBV-ACLF could induce the upregulated expression of BTLA on CD4[+] T cells. Moreover, we found that IL-6 and TNF-α, generally

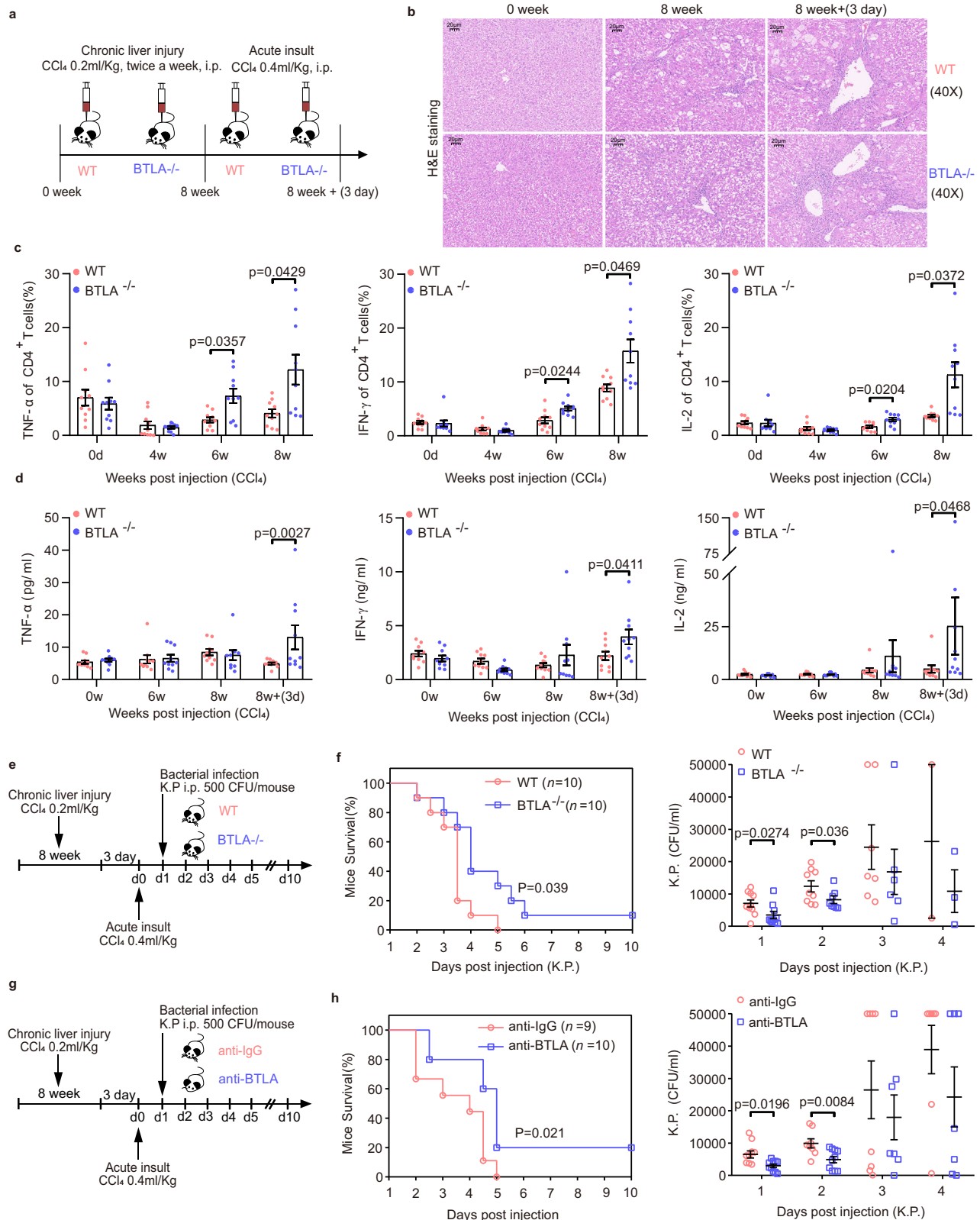

induced by LPS, may promote BTLA expression on CD4+ T cells via the STAT3 and NF-κB signaling, which explains why HBV-ACLF patients with coinfection have elevated BTLA levels.

BTLA has been shown to maintain T-cell tolerance in multiple ways. However, it is not clear whether it contributes to the induction of CD4+ T immune exhaustion in HBV-ACLF. In this study, utilizing an agonistic anti-BTLA monoclonal Ab, we found that crosslinked BTLA

markedly inhibits the expression of CD25, CD38, and CD69, the production of IFN-γ, IL-2, and TNF-α, and the proliferation of BTLA+CD4+ T cells while promoting the apoptosis of BTLA+CD4+ T cells. These results were confirmed on the CD4+ T cells from the peripheral blood of NC, CHB, and HBV-ACLF patients and were similar to those reported in a previous study[12]. However, the inhibitory effect mediated by the aforementioned anti-BTLA monoclonal Ab-mediated inhibiting effect

**Fig. 8 | BTLA contributed to immune exhaustion and increased mortality and infection rates in the ACLF model induced by CCl₄. a** Schematic diagram of ACLF induced by continuous injection of CCl₄ (0.2 mL/kg, twice a week) for 8 weeks and subsequent injection of a double dose of CCl₄ (0.4 mL/kg). **b** Hematoxylin-eosin diagram of liver pathology at baseline, 8 weeks, and 8 week + 3 days in WT ($n = 10$ mice) and BTLA$^{-/-}$ ($n = 10$ mice) C57BL/6 mice. **c** Cytokine (TNF-α, IL-2, and IFN-γ) production in CD4$^+$ T cells was higher in BTLA$^{-/-}$ mice ($n = 10$ mice) than in WT mice ($n = 10$ mice) at 6 and 8 weeks. **d** Cytokine (TNF-α, IL-2, and IFN-γ) levels were markedly increased in the plasma of BTLA$^{-/-}$ mice ($n = 10$ mice) compared to those in

WT mice ($n = 10$ mice) after a double dose of CCl₄ injection. **e, g** Schematic diagram of intraperitoneal (i.p.) K.P. and anti-BTLA antibodies. **f, h** Survival rate and K.P. load in peripheral blood of both BTLA$^{-/-}$ ($n = 10$ mice) and WT mice injected with anti-BTLA antibodies ($n = 10$ mice) were higher than those in the WT ($n = 10$ mice) and anti-IgG injection groups ($n = 9$ mice). Data were calculated as mean ± SEM (**c, d, f, h**), two-way ANOVA followed by Sidak's multiple-comparison test (**c, d**), Wilcoxon test (**f, h** right), log-rank test for survival study (**f, h** left). A two-sided $P < 0.05$ was considered significant.

could be reversed once BTLA expression on CD4$^+$ T cells was reduced by BTLA shRNA, while the secretion of serum cytokines and activation of CD4$^+$ T cells were increased in BTLA$^{-/-}$ ACLF mice, suggesting that BTLA may induce CD4$^+$T-cell exhaustion by inhibiting their activation and proliferation and the production of secretory cytokines, or by promoting the apoptosis of CD4$^+$ T cells. Previous studies have demonstrated an elevation in BTLA expression in a mouse model of hemorrhagic shock (Hem) following CLP, while anti-BTLA antibody (clone 6A6; reported to have the ability to neutralize BTLA signaling) treatment increased inflammatory cytokine storms, which in turn aggravated organ injury and mortality[38]. In contrast, in this study, the administration of a neutralizing anti-BTLA antibody effectively reduced bacterial burden and mortality in ACLF mice. The main reason for the different results is that Hem/CLP is an acute infection model, while the ACLF model induced by ConA and CCL4 in this study induced an acute injury on the basis of the chronic infection; in the former, an acute immune response occurred, while in the latter, continuous inflammation caused immune exhaustion. Knocking out or blocking the BTLA pathway reversed exhaustion and promoted pathogen clearance. Therefore, releasing the BTLA "brake" upon blocking its pathway activates the CD4$^+$ T cells and permanently changes the stable balance to eliminate pathogens, which may be applied to immunotherapy for HBV-ACLF.

Moreover, Murphy et al.[39] found that BTLA may mediate pro-survival signals under conditions of chronic stimulation in the non-irradiated parent-into-F1 and graft vs. host disease. The result indicates that BTLA may act not only as an inhibitor[40] but also as a provider of co-stimulatory signals that promote T-cell survival under some conditions[39]. Although it was found that BTLA contains a conserved intracellular tyrosine motif suggestive of a Grb-2 recruitment site, which can interact with Grb-2 and the p85 subunit of PI3K in vitro[41], the exact signaling pathway by which BTLA provides co-stimulatory signals is still unclear. We screened the signaling pathway through RNA-Seq and found that the PI3K-Akt pathway was the most probable candidate among the other five pathways. Additionally, the western blotting demonstrated that crosslinked BTLA promotes the phosphorylation of SHP-1/2, consistent with findings from previous studies[11,42], and also activates the PI3K-Akt-GSK-3β pathway in purified CD4$^+$ T cells. However, further investigations are necessary to verify these results.

In conclusion, our data demonstrate a clear upregulation of BTLA expression on CD4$^+$ T cells in the circulatory system and liver tissue of patients with HBV-ACLF, which strongly correlates with disease severity, prognosis, and impaired antimicrobial response. Functionally, crosslinking of BTLA was found to inhibit the activation, proliferation, and cytokine production of CD4$^+$ T cells, but it also promoted apoptosis in CD4$^+$ T cells, while a decreased BTLA expression via shRNA reversed the BTLA-mediated T-cell exhaustion. These may explain the T-cell exhaustion and susceptibility to infection. Mechanistically, the upregulation of BTLA on CD4$^+$ T cells induced by SIRS such as IL-6 and TNF-α suppresses signaling by phosphorylating SHP1/2 and, simultaneously, provides a pro-survival signal upon activating the PI3K-Akt signaling pathway. These results help elucidate the pathogenesis of HBV-ACLF, which may assist in identifying new drug targets.

## Methods

### Ethical approval

All studies were approved by the Ethics Committee of Huashan Hospital, affiliated with Fudan University (No. KY2021-652), and Biomedical Research Affiliated to Fujian Medical University (No. 2014-87). Written consent was obtained from healthy individuals and patients with CHB and HBV-ACLF. The protocols used for animal experimentation were approved by the Animal Ethics Committee of School of Basic Medical Sciences, Fudan University (No. 20171561A701).

### Patients

Blood samples were collected and PBMC were isolated from 104 patients with CHB and 101 with severe liver injury (TBil ≥ 5 mg/dL and INR ≥ 1.5) within 24 h of admission to the Department of Infectious Diseases, Huashan Hospital affiliated with Fudan University, and the First Hospital of Quanzhou affiliated with Fujian Medical University from January 2014 to September 2018. Ninety healthy individuals were enrolled as normal controls. Liver tissues were obtained from five patients with hemangioma, five with CHB, and seven with HBV-ACLF from the Department of Hepatobiliary Surgery, Huashan Hospital, affiliated with Fudan University. All patients were diagnosed according to previously described criteria[24,43,44]. Briefly, we recruited CHB patients, including patients with positive hepatitis B surface antigen (HBsAg) detection for more than 6 months, hepatitis-related clinical manifestations, histological confirmation of hepatitis, and abnormal alanine aminotransferase (ALT) levels (≥40 U/L). Patients with HBV-ACLF were defined as follows: TBil≥ 12 mg/dL and INR ≥1.5, regardless of the presence of cirrhosis, complicated by hepatic and/or extra-hepatic organ failure. Among 101 patients with severe liver injury (TBil ≥ 5 mg/dL and INR ≥1.5), 71 patients with HBV-ACLF were enrolled in this study (Supplementary Table 1). Patients were excluded if they were co-infected with any other virus (e.g., Hepatitis A, C, and D), had drug- or alcohol-induced liver diseases, had hepatocellular carcinoma or other tumors, or were treated with immunosuppressive drugs.

### Hematologic, biochemical, and virological parameters

Albumin (ALB), total bilirubin (TBil), direct bilirubin (DBil), alanine transaminase (ALT), aspartate aminotransferase (AST), cholinesterase (CHE), white blood cell (WBC) count, neutrophil count, prothrombin time (PT), international normalized ratio (INR), C-reactive protein (CRP), procalcitonin (PCT), hepatitis B virus surface antigen (HBsAg), hepatitis B virus e antigen (HBeAg), and hepatitis B virus DNA were documented at the time of admission.

### Cell isolation and sorting

Peripheral blood mononuclear cells (PBMC) were isolated from ethylenediamine tetra-acetic acid-anticoagulant venous blood using Ficoll−Hypaque density gradient centrifugation (Cedarlane Laboratories) as previously described[45]. Fresh liver tissues were cut out. Next, the single-cell suspension was collected and washed with phosphate-buffered saline. Afterwards, the cell suspension was resuspended in 40% Percoll (Sigma-Aldrich), gently overlaid onto 70% Percoll, and centrifuged for 20 min at 2000 rpm. Liver-infiltrating lymphocytes were collected from the interface[46]. Peripheral blood CD4$^+$ T cells were purified using magnetic beads (Miltenyi Biotec) at a purity level of

≥90%. BTLA$^+$CD4$^+$ T and BTLA$^-$CD4$^+$ T cells were sorted using a MoFlo XDP cell sorter (Beckman Coulter) with a purity >95%.

## Flow cytometry

To detect the expression and distribution of BTLA in liver-infiltrating lymphocytes and circulating CD4$^+$ T-cell subtypes, as well as the phenotypic differentiation of BTLA$^+$CD4$^+$ T cells, PBMC from NC, CHB, and HBV-ACLF patients were labeled with the following monoclonal antibodies (mAbs): APC anti-human CD3 (Biolegend), BV510™ anti-human CD4 (BD Biosciences), PE/Cy7 anti-human CD8 (BD Biosciences), Percp/Cy5.5 anti-human BTLA (Biolegend), FITC anti-human CD27 (Biolegend), APC/Cy7 anti-human CD45RA (Biolegend), or APC/Fire™ 750 anti-human CD45 (Biolegend). Detailed information about all antibodies is shown in Supplementary Table 2.

To investigate the expression of BTLA in the circulation, CD4$^+$ T-cell subgroups and PBMC from NC, CHB and HBV-ACLF patients were labeled with the following mAbs: APC anti-human CD3 (Biolegend), Brilliant Violet 510™ anti-human CD4 (BD Biosciences), Percp/Cy5.5 anti-human BTLA (Biolegend), BV510™ anti-human CCR4 (Biolegend), APC/Cy7 anti-human CCR6 (Biolegend), PE anti-human CCR10 (Biolegend), BV421™ anti-human CXCR3 (Biolegend), and PE/Cy7 anti-human CXCR5 (Biolegend). Data acquisition and analysis were performed using MoFlo XDP (Beckman, USA).

For characterizing the phenotype of BTLA$^+$CD4$^+$ T cells, PBMC from NC, CHB, and HBV-ACLF patients were labeled with the following mAbs: APC AF750 anti-human CD3 (Beckman), ECD anti-human CD4 (Beckman), FITC anti-human CCR5 (Biolegend), PE anti-human BTLA (Biolegend), PC5 anti-human CD127 (Beckman), PC7 anti-human CD64 (Beckman), APC anti-human CD25 (Beckman), APC A700 anti-human CD7 (Beckman), and PB anti-human CD57 (Beckman). Controls for each experiment included unstained cells and cells that were single-stained for surface markers or intracellular proteins.

## Immunofluorescence double-staining

For immunofluorescence double-staining, sections were deparaffinized with xylene and ethyl alcohol and quickly rinsed with TBST buffer for 5 min. After blocking with 5% normal serum-TBST for 1 h, sections were incubated with mouse monoclonal anti-CD4 (eBioscience, USA), anti-BTLA (Abcam, USA) antibodies at 4 °C overnight. After washing with TBST (5 min, three times), sections were incubated with donkey anti-rabbit Alexa 488 and donkey anti-mouse Alexa 555 (Jackson ImmunoResearch, West Grove, PA, USA) in the dark for 30 min. Finally, sections were incubated with 1 µg/mL DAPI (Sigma, CA, USA) for 10 min to stain nuclei. The results were analyzed using an inverted Eclipse Ti-S microscope (Nikon, Japan). Liver sections were stained with hematoxylin & eosin (H&E).

## Determination of cytokine/chemokine concentrations

Levels of pro-inflammatory cytokines (IL-1β, IL-6, TNF-α, IL-4, IL-17A, IFN-γ, and IL-12), anti-inflammatory cytokines (IL-10), and other chemokines (GM-CSF, FGF2, MDC, IP-10, MCP-1, and MIP1-α) in the plasma of patients or cytokines (TNF-α, IL-2, IL-10, and IFN-γ) in the plasma of mice were determined using the Luminex 200 multiplexing instrument (EMD Millipore, USA). Patients' plasma levels of IL-37, IL-21, and IL-22 were measured using an ELISA kit (RayBiotech Inc, Norcross, GA, USA) according to the manufacturer's protocol.

## Induction of BTLA expression on CD4$^+$ T cells in vitro

PBMC from healthy donors were cultured in complete medium (RPMI 1640 supplemented with 10% heat-inactivated FBS, 100 µ/mL penicillin and 100 µg/mL streptomycin [Life Technologies]) at a concentration of $1 \times 10^6$ cells/mL for 3, 5, and 7 days with the plasma of HBV-ACLF and NC, or various concentrations (10–50 ng/mL) of recombinant human (rh) interleukin rhIL-1β (PeproTech), rhIL-6 (PeproTech), rhIL-10 (PeproTech), rhIL-22 (PeproTech), rhIL-37 (R&D), and tumor necrosis

factor (rhTNF-α, PeproTech), or 10 µg/mL neutralizing antibody of anti-IL-6 (PeproTech), anti-TNF-α (PeproTech), and anti- IL-1β (PeproTech). The medium and cytokines were refreshed every other day. The expression of BTLA in CD4$^+$ T cells was analyzed using flow cytometry.

PBMC from healthy donors were cultured at a concentration of $1 \times 10^6$ cells/mL in complete medium and were treated for 3 days in the following groupings: (a) control; (b) stimulant alone: rhIL-6 alone (50 ng/mL), or rhTNF-α (50 ng/mL); (c) inhibitor alone: anti-stat3 (SH-4-54, 50 µmol/L, Selleck) or the anti-NF- kappa (QNZ, 20 µmol/L, Selleck) or Polymyxin B (PXB, 10 µg/mL, Sigma-Aldrich); and (d) stimulant plus inhibitors. All experiments were performed at least in duplicate.

## Quantification of BTLA expression and bacterial 16 S rDNA using RT-PCR

To measure mRNA expression of *BTLA* and *Stat3*, samples were obtained from PBMC of study subjects or from PBMC treated by recombinant human cytokines. Total RNA was extracted by Trizol (Invitrogen) and reverse-transcribed into complementary DNA (cDNA) using a Fast Quant RT Kit (Tiangen). Expression levels were quantified via real-time PCR (SYBR® Premix Ex Taq II, Tiangen) according to the manufacturer's instructions. The procedure for the detection of bacterial 16S rDNA in whole blood of mouse models was carried out as previously reported[9]. Specific primers for *BTLA, Stat3, 16s, and β-actin* were designed (Supplementary Table 3). Values were calculated as the difference in cycle threshold (Ct) values normalized to those of mRNA encoding β-actin for each sample using the following formula: relative RNA expression = $2^{-\Delta Ct} \times 10^3$.

## Electroporation with BTLA shRNA

PBMC from healthy donors were divided into three groups: BTLA shRNA, control shRNA, and GFP. Cell concentrations were adjusted to $5–10 \times 10^6$ using RPM-1640 (Corning, USA). The required numbers of cells were pelleted and the supernatant was completely discarded. Next, the PBMC were carefully resuspended with 2 µL pmaxGFP® Vector, plus 2 µL BTLA shRNA or 2 µL control shRNA in 100 µL room-temperature Nucleofector® Solution, which was mixed with transfection solution and complete supplement solution at a ratio of 4.5:1. The PBMC suspension was transferred into a certified cuvette. Next, 500 µL of pre-preheated free serum-X-VIVO medium was added to the three cuvettes, and the PBMC were gently resuspended and transferred to 12-well plates (final volume of each 2 mL). The PBMCs were incubated at 37 °C in 5% CO$_2$ for 3 days, and only PBMC with decreased expression of BTLA were used in subsequent experiments.

## T-cell activation assay

Untreated control PBMC or shRNA-treated PBMCs from healthy donors were stimulated with anti-CD3/CD28 antibodies (1 mg/mL; eBioscience, San Diego, CA) in medium with plate-coated anti-BTLA (10 µg/mL; eBioscience, San Diego, CA) or PBS (Corning, USA) for 1 day. The cells were then selected, and the expression of CD25, CD38, CD69, PD-1, CTLA-4, TIM-3, and TIGIT was analyzed via flow cytometry. All antibodies were purchased from BD Biosciences (San Jose, CA, USA).

## Proliferation assay

Untreated control PBMC, PBMC knocked down by shRNA, or sorted BTLA$^+$CD4$^+$ T cells and BTLA$^-$CD4$^+$T cells from healthy donors were labeled with 2.5 mM carboxyfluorescein succinimidylester (CFSE) (Thermo, USA) for 15 min at 37 °C. CFSE-labeled cells were cultured in 96-well plates with anti-CD3/CD28 antibodies (1 mg/mL; eBioscience, San Diego, CA) in medium, with plate-coated anti-BTLA (10 µg/mL; eBioscience, San Diego, CA) or PBS for 5 days. The cells were then selected and the CFSE dilution was evaluated by flow cytometry.

### Annexin V-apoptosis assay

Untreated control PBMC or PBMC knocked down by shRNA from healthy donors were treated in 96-well plates with plate-coated anti-BTLA (10 μg/mL; eBioscience, San Diego, CA) or PBS for 1 d, washed with PBS followed by co-staining with Annexin V and PI (BD Bioscience). The apoptosis rate was examined using flow cytometry.

### Analysis of cytokine production

Untreated PBMC, PBMC knocked down by shRNA, or sorted BTLA$^+$CD4$^+$ T and BTLA$^-$CD4$^+$ T cells from healthy donors were stimulated in 96-well plates with PMA/ionomycin/BFA (sigma-Aldrich), with or without plate-coated anti-BTLA (10 μg/mL; MIH26, eBioscience, San Diego, CA) or PBS for 5 h. The cells were labeled with surface antibodies (APC anti-human CD3 and FITC anti-human CD4) and then fixed and permeabilized using the Cytofix/Cytoperm Plus kit (BD Biosciences). Lastly the cells were stained with intracellular antibodies (BV421 anti-human IFN-γ, PE anti-human IL-2, and APC/Cy7 anti-human TNF-α). All antibodies were purchased from BD Bioscience (San Jose, CA, USA).

### Whole-transcriptome library preparation and sequencing

PBMC from 3 NC, 3 CHB patients, and 3 HBV-ACLF patients were treated in 96-well plates with plate-coated anti-BTLA (10 μg/mL; eBioscience, San Diego, CA) or PBS for 12 h. BTLA$^+$CD4$^+$ T and BTLA$^-$CD4$^+$T cells were sorted with a MoFlo XDP cell sorter (Beckman Coulter) with a purity of >95%. Total RNA from PBMC or sorted T cells was extracted using the miRNeasy Mini Kit (Qiagen) following the manufacturer's instructions, and the RNA integrity number (RIN) was obtained using an Agilent Bioanalyzer 2100 (Agilent technologies, CA, USA). Qualified total RNA was further purified by the RNA Clean XP Kit (Beckman Coulter, Inc. CA, USA) and the RNase-Free DNase set (QIAGEN, GmBH, Germany). Ribosomal RNA (rRNA) was removed using Agencourt® AMPure XP Beads (Beckman Coulter, Inc. CA, USA) according to the manufacturer's instructions. Subsequently, strand-specific sequencing libraries were generated using Superscript II Reverse Transcriptase (Invitrogen, USA). Library quality was assessed using the Agilent High Sensitivity DNA Kit (Agilent technologies, CA, USA). RNA sequencing (RNA-Seq) was performed on an Illumina Hiseq 2000 platform, and 100 bp paired-end reads were generated according to Illumina's protocol.

### Protein isolation and western blotting

Purified CD4$^+$ T cells from healthy donors were treated in 96-well plates with plate-coated anti-BTLA (10 μg/mL; eBioscience, San Diego, CA) or PBS for 12 h and scraped and transferred into Eppendorf tubes, which were then centrifuged at 3000 rpm for 5 min; the cells precipitated at the bottom of the Eppendorf tubes. The cells were then lysed in radioimmunoprecipitation assay (Absin Bioscience, Beijing) lysis buffer containing phenylmethanesulfonylfluoride. Protein concentrations were measured using a bicinchoninic acid kit (Sigma-Aldrich, USA). The samples were separated using SDS-PAGE and then transferred onto a polyvinylidene difluoride (PVDF) membrane (Millipore, USA). The PVDF membrane was blocked with 5% nonfat milk and incubated with the following primary antibodies: PI3K, phospho-PI3K, Akt, phospho-Akt, phospho-GSK-3β, CREB, phospho-CREB, and phospho-SHP1/SHP2 (Cell Signaling Technology, Beverly, Mass). GAPDH (Biodesign International, Saco, Maine) was used as the loading control. The blots were incubated with the horseradish peroxidase-conjugated secondary antibodies (Santa Cruz Biotechnology) for 1 h at room temperature.

### Animal model

Male wild-type (WT, Cat. NO. SM-001) and BTLA$^{-/-}$ C57BL/6 mice (Cat. NO. NM-KO-18051), both 8 weeks old and weighing 20 ± 1 g, were purchased from the Shanghai Model Organisms Center, Inc.

(Shanghai, China). They were bred in a specific pathogen-free barrier facility (temperature: 20–26 °C, humidity: 40-70, dark/photoperiod: 12 h/12 h). For the ACLF model, Concanavalin A (ConA, 8 mg/kg, Sigma-Aldrich, USA) in 0.9% saline was injected into the retrobulbar angular vein every 2 days for 8 times. A septicemia model induced by cecal ligation and puncture (CLP) was established as previously described[47]. Mice were anesthetized with ketamine and xylazine, and a 1-cm midline incision was made in the abdomen to expose the cecum. The cecum was ligated from the end and then punctured twice with a 21-gauge needles. The abdominal incision was closed using two or three surgical wound clips, and the mice were resuscitated using 1 mL of sterile saline. After the observation period, the remaining mice were placed in the original cage box and sent to a transparent euthanasia container, where the $CO_2$ flow rate was 40 L/min, and it was closed after 3 min. when all mice had stopped breathing and lost eye color, they were removed from the euthanasia box, and when once again confirmed that the mice had stopped heartbeat and breathing, stiff body and dilated pupils, they were sent to the refrigerator in the waste room for storage.

For another ACLF model, carbon tetrachloride (CCl$_4$, 0.2 mL/mg) was injected into the abdomen of WT and BTLA$^{-/-}$ mice twice a week for 8 weeks, followed by a double dose injection of CCl$_4$ (0.4 mL/kg), an intraperitoneal injection of *Klebsiella pneumoniae* (K.P.) strain 43816[29], and (or) plus Ultra-LEAF™ purified anti-mouse BTLA antibody (1 μg/mL, BioLegend) or its isotype IgG control for WT mice. The peripheral blood and liver tissues were harvested after the indicated days.

### Quantification analysis of liver pathology

The pathological images were scanned by a panoramic section scanner (PANNORAMIC DESK/MIDI/250/1000, 3DHISTECH. Ltd). Histological grading for liver necroinflammation and fibrosis was performed according to International Harmonization of Nomenclature and Diagnostic Criteria for lesions in rats and mice (INHAND)[48] by specialized pathologists. A grade of 0 zero (normal limits) indicates that inflammation and fibrosis are within the normal range; 1 (slight), that is, just beyond the normal range; 2 (mild), the lesion is detectable but not serious; 3 (moderate), the lesions are obvious and more severe; and 4 (severe), the lesions occupy the entire tissue and organ.

### Statistical analyses

GraphPad Prism 7.0 (GraphPad Software Inc., San Diego, California, USA) software was used. Data were calculated as medians and interquartile ranges, or mean ± standard error of the mean (SEM) unless otherwise specified. Comparisons of data between or among groups were performed using the Wilcoxon test, the Mann–Whitney $U$ test, Kruskal-Wallis H test followed by Dunn's multiple comparison test, one-way ANOVA followed by Tukey's and Dunnett's multiple comparison test, or Two-way ANOVA followed by Sidak's multiple-comparison test. A correlation analysis was performed using the Spearman tests. A survival study was performed using log-rank test. A two-tailed $P < 0.05$ was considered significant.

### Reporting summary

Further information on research design is available in the Nature Portfolio Reporting Summary linked to this article.

## Data availability

The RNAseq data generated in this study have been deposited in the NCBI under accession code GSE248217. All data are included in the Supplementary Information or available from the authors upon reasonable requests, as are unique reagents used in this article. Source data are provided with this paper.

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

## Acknowledgements

We thank Xiaogang Xiang and his team for their guidance in building the mouse model. We also thank Gan Zhao and Bin Wang for their excellent technical assistance and critical reading. This study was supported by the National Natural Science Foundation of China (82370604, 82372237, 82160361, and 82003864), the Major Science and Technology Special Project of China (2017ZX10202202, 2017ZX10202203-007), the Natural Science Foundation of Fujian Province (2023J01239, 2019J01593), the Shanghai Municipal Science and Technology Major Project (ZD2021CY001), and the Shanghai Key Clinical Specialty Construction Program (ZK2019B24).

## Author contributions

Study concept and design: XPY, FFY, ZLS, YZ, JS, SYZ, CQ, JQX, RCM, ZJS, and JMZ. Acquisition, analysis, and interpretation of data: XPY, FFY, ZLS, YZ, CQ, YJZ, JQX, and JMZ. Material support: SYZ, YJZ, WDZ, SHY, DWZ, JFL, MQZ, XYZ, JWW, ZXM, HXZ, MLS, and BL. Drafting of the manuscript: XPY, FFY, ZLS, YZ, and RCM. Critical manuscript revision for important intellectual content: XPY, FFY, ZLS, RCM, ZJS, and JMZ. Obtained funding: XPY, ZLS, ZJS, and JMZ.

## Competing interests

The authors declare no competing interests.
