## [Peer Review File · Nature Communications]

REVIEWER COMMENTS

Reviewer #1 (expertise in immune checkpoint blockade and liver Immunology):

In this manuscript, Yu, Yang et al. measured BTLA expression in blood T cells from HBV-ACLF patients (compared to control groups), identified increased BTLA levels in CD4+ T cells in HBV-ACLF, and therefore went on to study the role of BTLA in regulating CD4+ T cell activation/function. They found that BTLA in CD4+ correlates with disease severity and inflammation; particularly, it correlated with patients' plasma TNF- α and IL-6 levels, so they performed a series of in vitro work to mechanistically link these cytokines to BTLA upregulation in HBV-ACLF; however, blockade of plasma TNF- α and IL-6 was not interrogated here. Moreover, they elegantly show how BTLA agonism (mAb) suppresses CD4+ T, and that shRNA-mediated reduction of BTLA expression can reverse its effects. Finally, they used a murine model of ConA-induced liver injury, compared WT vs BTLA KO mice, to interrogate BTLA's role in vivo. Overall, the study is interesting, includes proper scientific methods and approaches. There are areas to improve within the manuscript and pursue further mechanistically, both in vitro and in vivo. Please see my comments below.

- Minor comments -

line:

42: "contrastingly" is misused, the two points compared are concordant to one another

113: "interesting"

119: ILC, acronym / Fig. 1C: liver-infiltrating or intrahepatic lymphocytic cells (ILC) ?

203: the-6, typo (rhIL-6)

206: dose-dependent (*manner*, missing)

251: 200 nm, typo (nM)

270: "Subsequently, through flow cytometry, western blotting", is confusing as it's WB data

In the Introduction or Discussion section, the authors could include/discuss few references re. BTLA blockade in past/current clinical trials in inflammatory pathologies.

- Major comments -

The ACLF diagnosis criteria need clarification (e.g., no mention of extra-hepatic organ failure; differentiation between ACLF and AD?)

Fig. 1D (lines 128-130): no quantitative data of HVEM expression of CD80/D86+ DC (% / MFI) in different cohorts (NC vs CHB vs HBV-ACLF) have been provided, only t-SNE plots; this is important re. HVEM-BTLA axis. Is HVEM expression altered in other APCs e.g., monocytes?

Fig. 3D: BTLA expression (%) in CD4+ T cells do not seem majorly reduced; the significance shown here is probably due to one sample only. What about BTLA expression at MFI level?

Fig. 3: The authors demonstrate that treatment of NC PBMCs with HBV-ACLF (vs CHB or NC) plasma induces BTLA upregulation in CD4+ T cells. However, this experimental set up doesn't show plasma's direct effects on CD4+ T cells (e.g., it could be immune cell-mediated effects on CD4+); this could be further addressed by culturing NC-derived CD4+ T cells with plasma. Given the CD4+ BTLA's correlation with plasma IL-6 and TNF- α , and by performing cell culture with rhIL-6 or rhTNF- α , the authors propose that these cytokines mechanistically induce BTLA expression in CD4+ cells in HBV-ACLF. To support this stronger, the authors should explore in vitro the blockade of IL-6 or TNF- α in plasma (e.g., anti-IL-6 or anti-TNF- α mAb), using both PBMCs and isolated CD4+ T cells from NC. Also, these two cytokines drive Th1; one would wonder if BTLA is Th1 related? e.g., same trends in IL-1 β treatment (but plateaued).

Fig. 4: In addition to activation markers (e.g., CD25, CD69) or cytokines screened, do CD4+ T cells differentially express immune checkpoints (e.g., PD-1, CTLA-4, TIGIT, Tim3) following BTLA

agonism? Also, in (Fig. 4C) the CFSE analysis needs clarification; how is it calculated, what is "CFSE %"

Fig. 5D (lines 256-257): BTLA shRNA-treated CD4+T cells tended to show higher apoptosis levels than the control shRNA-treated CD4+T cells; data show the opposite, needs correction.

Fig. 5E: CD69 expression is at 60% on culture Day 3; it should drop after 12 hours, this needs explanation; the authors could provide some representative flow cytometry plots for the stains assessed e.g., CD69 (same for anti-mouse CD69).

Fig. 7: The authors used a ConA (8 mg/kg) murine model to mimic ACLF and compare WT vs BTLA KO animals. However, the model is poorly characterised here:

1) histological stains: a) only representative H&E stains shown; what about quantitative data? the areas of liver damage/necrosis (this can be quantified as % of area) or the inflammatory cell infiltrate (histopathological score), b) PicroSiriusRed (PSR) stains re. liver fibrosis have not been included; this could be measured as area of collagen (%). How do these differ in WT vs BTLA mice? Other indices of liver injury should be measured e.g., serum ALT, AST levels.

2) Re. flow cytometry: representative gating strategies to stain and identify mouse liver CD4+ T cells and measure BTLA are not included; similarly, to what was explored in human liver, is BTLA differentially expressed in mouse liver CD8+ or NK cells following ConA-induced injury? Representative flow cytometry stains of CD25, CD38, CD69 and cytokines should be included (probably in suppl. material).

3) Given that following CLP, the mortality rate and bacDNA levels were decreased in BTLA KO mice (compared to WT) the therapeutic in vivo blockade of BTLA (e.g., administration of anti-mouse BTLA mAb following induction ConA-induced injury, or prior to liver injury) should be also investigated to assess its clinical use potential (translation into future therapy).

- The CLP sepsis after ConA-induced liver injury should be studied in more detail (it could be Fig. 8 on its own). In addition to mortality and bacDNA, the authors should also examine tissue histology (e.g., liver, kidney, spleen, lungs), serum cytokines (pro- or anti-inflammatory) and indices of liver injury/damage (H&E/PSR stains, serum ALT/AST).

Importantly, the ConA (8 mg/kg) model is different, modified to mimic ACLF in mice, to the one described in text (ref. 24: ConA in rats), therefore it's not well-established. A well-characterised murine ACLF model in the field is the CCl₄-induced liver fibrosis (6-10w) model in combination with i.p. bacterial infection (e.g., *E. coli*) or i.p. LPS (PMID: 31987990 and PMID: 31786256). This model hasn't been examined in this manuscript.

Reviewer #2 (expertise in liver Immunology):

In their manuscript Yu et al. describe the upregulation of BTLA on effector T cells from patients with HBV complicated by ACLF in association with infectious complications and poor survival. In vitro and in a ko mouse model they demonstrate an association of BTLA expression on CD4 T cells and T cell exhaustion markers. In a BTLA4 ko mouse model they also reveal a protective effect following cecal ligation. While the manuscript details a number of new and potentially interesting observations, and may suggest BTLA as a biomarker on T cells in the context of HBV-ACLF, questions remain in regards to the mechanism and its potential translation.

Comments:

- CD4 T cell responses play an important role in HBV infection. Moreover, ACLF is a highly complex systemic condition. In this manuscript it generally remains unsolved whether the phenomenon of BTLA expression and T cell exhaustion described was due to HBV, HBV induced cirrhosis, ACLF or superimposed bacterial infection? Does it also occur in ACLF of other aetiology? Or in intensive care patients with other infections? What of these components exactly leads to BTLA expression presumably mediated by TNF and IL6?

- In Figure 1 E-H (Results page 7/8, line 133 onwards) it remains unclear whether CD4+ T cells here are derived from the blood or the liver.

- Figure 1 H/ Supplementary 5E: what is the definition of "good" and "poor" prognosis?

- Overall it appears that the BTLA+- and BTLA-CD4+ T cells are very similar, what do the authors

think is the biological significance? Figure 2B/C: Are the data obtained from trace RNAseq and flow cytometry contradictory? At least differences in CD7, CD127, CD25 seem to be detected only in flow cytometry. Figure 2E does not seem to reveal significant differences in the studied population of ACLF.

- The upregulation of diverse cytokines in ACLF (Figure 3B) has been repeatedly described before.
- Concanavalin A (ConA) is a model for acute immune-mediated hepatitis such as viral or autoimmune hepatitis. Apart from its initial description in the cited reference (Ref 24), it has not become a recognised ACLF model (Ref doi: 10.14218/JCTH.2022.00086). Also blocking BTLA has been shown beneficial in another mouse model of acute hepatitis (MHV-3, Ref 14). It is unclear why the authors did not choose a more established animal model to simulate ACLF?
- Discussion line 318: the Ref Chen et al seems relevant but is missing in the list of publications.
- Methods: 104 CHB patients and 101 severe liver injury – 71 due to HBV – were the other liver injury patients also analysed? Please state this accurately. Were the 9 controls age matched?

Minor comments:

- The abbreviation ILC is reserved for "innate lymphoid cells" and therefore should not be used to abbreviate "intrahepatic lymphocytes cells". Also in Figure 1C these cells are then termed "liver infiltrating lymphocytes". Consistency with terms is missing.
- Overall the language needs review by a native speaker.
- Page 6, line 113 "interestingly, ..."
- page 8, line 147 – what are "coinfection patients" HBV/HDV coinfecting have been excluded?! Or bacterial infection superimposed on HBV-ACLF? Please specify.
- Page 10, line 203: typo "the-6"; wording line 207.
- Suppl Fig8 G,H,I: anti-BTLA should always be displayed on the same side.

Reviewer #3 (expertise in hepatology and liver failure):

In their paper Yu et al investigated the relevance of the B- and T-lymphocyte attenuator (BTLA) in T cell immune response with regard to infections and mortality risk in patients with HBV-related ACLF. The analyses include characterization of PBMCs from 175 chronic hepatitis B patients of whom 71 were defined to have ACLF according to COSSH ACLF criteria and 90 healthy controls.

The main findings of this study are: A) BTLA is significantly upregulated in peripheral and intrahepatic CD4+ T cells of patients with HBV-related ACLF compared to CHB and healthy controls. Accordingly, the expression of BTLA ligand HVEM is upregulated on DCs of HBV-ACLF patients. B) Upregulation of BTLA expression in CD4+T cells correlates with disease progression, prognosis, and infectious complications in HBV-ACLF patients. C) Further characterization of BTLA+CD4+ T cells revealed that BTLA is expressed on CD4+ T cells during inflammatory states, inhibits T cell activation and cytokine production but promotes CD4+ T cell apoptosis and exhaustion. D) In an in vivo ACLF model immune exhaustion was seen in BTLA ko but not in WT mice.

In summary, the study aims at identifying functional and prognostic relevance of BTLA expression on CD4 T cells in patients with HBV-ACLF and to gain insights in its impact on T cell biology, especially T cell exhaustion in ACLF and systemic infection. These authors present a methodologically sound manuscript with interesting mechanistic insights in the regulation and function of BTLA expression on CD4 T cells in HBV-ACLF.

Nevertheless, the following aspects indicate necessity of major revision of the manuscript:

Major aspects:

1. ACLF classification: ACLF scoring of included patients was performed using the COSSH-ACLF score. The classification is different to the EF CLIF classification regarding two main aspects: 1) In contrast to the EASL-CLIF ACLF score, the COSSH score includes patients with HBV-related ACLF with AND without liver cirrhosis. The authors explicitly state here that patients were included "regardless of presence of cirrhosis" (l. 408). That means, that within the patients' "HBV-ACLF" cohort might also include patients with acute liver failure. Therefore, the described results for the role of BTLA are not specific for AC(!)LF. 2) Failure of each organ system is assessed using the

CLIF-C Organ Failure Score in both cases, however and in contrast to the EASL-CLIF score, according to the COSSH-ACLF patients with single liver failure and either INR>1.5 OR creatinine ranging from 1.5-1.9 are classified as ACLF grade 1 but would not meet the criteria for ACLF I° according to the EF-CLIF classification. The authors should discuss what impact the classification using the COSSH-ACLF score has in comparison to other internationally recognized score (EASL-CLIF, NACSELD) regarding the results of their work and whether the functional and prognostic relevance of BTLA is valid also if the included patients are scored by EF-CLIF ACLF score.

2. Generality of results: The authors describe a prognostic relevance of BTLA in HBV-ACLF. However, as the main findings of the study relate to CD4 T cells, the findings are not specific for HBV-related liver diseases or viral immune defense. Therefore, the authors should investigate whether the upregulation of BTLA on CD4+ cells is also present within other ACLF etiologies to demonstrate the generality of the biological role of BTLA in ACLF. Therefore, a second cohort of patients with another etiology of cirrhosis, i.e. alcoholic cirrhosis, have to be included in the analysis and findings need to be reproduced here.

3. Methodology: Please also provide MFI data for HVEM expression on CD80/86+ DCs as complementation for subfigure 1D.

4. CD25: Figure 2C shows an upregulation of CD25 in MFI on CD4+ BTLA+ T cells in ACLF patients. The authors argue that the BTLA expression is accompanied by an upregulation of CD25 as a marker for T cell activation. In contrast, in the in vivo model an upregulation of CD25 is seen on CD4+ T cells in BTLA ko(!) mice. How do these findings fit together? This should be further discussed.

Minor aspects:

1. PBMC sampling: What was the time point of blood draw for PBMC isolation? Was this hospital admission and was there a maximum of hospital admission duration (i.e. within first 72h of admission) until when the PBMC had to be collected? If not sampling error could occur due to therapeutic measures and already resolved ACLF, especially in low grade ACLF patients. The authors should comment on this.

2. Line 152: do you mean survivors and non-survivors by "good prognosis" and "poor prognosis"? In your previous publication you are referring to (#22) the groups are called survivors and non-survivors. Please stick to that labeling if these are the same groups, if not, please state in the text in more detail how these groups are characterized and how these groups then refer to your previous publication

3. Please add the exact number of ACLF patients (71) to the results part.

4. Please correct "interesting" in line 113 to "interestingly"

RESPONSE TO REVIEWERS' COMMENTS

Reviewer: 1# (expertise in immune checkpoint blockade and liver Immunology):

Minor comments

1. line: 42: “*contrastingly*” is misused, the two points compared are concordant to one another.

Authors' response: Thank you for pointing out this error; it has been corrected to “meanwhile” (line 46).

2. line: 113: “*interesting*”

Authors' response: Thank you for pointing out this error; it has been corrected to “Notably” (line 134).

3. line: 119: *ILC, acronym / Fig. 1C: liver-infiltrating or intrahepatic lymphocytic cells (ILC)?*

Authors' response: Thank you for the reminder. The “LIL” acronym has been defined as “liver-infiltrating lymphocytes (LIL)” (lines 140–141).

4. line:203: *the-6, typo (rhIL-6)*

Authors' response: Thank you for pointing this out; it has been corrected (line 229).

5. line:206: *dose-dependent (*manner*, missing)*

Authors' response: Thank you for pointing out this error; it has been corrected (line 231).

6. line:251: *200 nm, typo (nM)*

Authors' response: Thank you. The typo has been corrected to “200 nM” (line 289).

7. line:270: “*Subsequently, through flow cytometry, western blotting*”, is confusing as it's WB data

Authors' response: Thank you for your careful review of our manuscript. This error has been corrected to “Subsequently, through western blotting, it was determined that...” (lines 310–311).

8. *In the Introduction or Discussion section, the authors could include/discuss few references re. BTLA blockade in past/current clinical trials in inflammatory pathologies*

Authors' response: We thank you for this insightful suggestion.

We have added two references to the introduction of the manuscript, as follows: In addition, BTLA-deficient mice have exhibited enhanced pathogen clearance compared to wild-type (WT) mice in the early phase of infection, while agonistic anti-BTLA antibodies have been shown to rescue mice from lipopolysaccharide (LPS)-induced endotoxic shock. These findings suggest that BTLA also plays a crucial role in the immune response against infectious pathogens^{15, 16} (lines 91–96).

We have also added a paragraph to the discussion section, as follows: Previous studies have demonstrated an elevation in BTLA expression in a mouse model of hemorrhagic shock (Hem) following CLP, while anti-BTLA antibody (clone 6A6; reported to have the ability to neutralize BTLA signaling) treatment increased inflammatory cytokine storms, which in turn aggravated organ injury and mortality³⁸. In contrast, in this study, the administration of a neutralizing anti-BTLA antibody effectively reduced bacterial burden and mortality in ACLF mice. The main

reason for the different results is that Hem/CLP is an acute infection model, while the ACLF model induced by ConA and CCL4 in this study induced an acute injury on the basis of the chronic infection; in the former, an acute immune response occurred, while in the latter, continuous inflammation caused immune exhaustion. Knocking out or blocking the BTLA pathway reversed exhaustion and promoted pathogen clearance (lines 433–444).

Major comments

1. The ACLF diagnosis criteria need clarification (e.g., no mention of extra hepatic organ failure; differentiation between ACLF and AD?)

Authors' response: Thank you for pointing out this important issue. HBV-ACLF was defined as follows: TBil \geq 12 mg/dL and INR \geq 1.5, regardless of the presence of cirrhosis, complicated by hepatic and/or extrahepatic organ failure⁴. Acute decompensation of cirrhosis (AD) is characterized by ascites, hepatic encephalopathy, upper gastrointestinal hemorrhage, and bacterial infection.

ACLF includes non-cirrhosis, compensatory cirrhosis, and decompensated cirrhosis. Therefore, in terms of scope, ACLF is broader, and AD represents only a small category of ACLF. In this study, only 57 of the 71 HBV-ACLF patients were AD patients. Moreover, although most AD symptoms are consistent with those of ACLF, there are still a small number of AD patients whose TBil and INR levels do not meet the diagnostic criteria of ACLF.

2. Fig. 1D (lines 128-130): no quantitative data of HVEM expression of CD80/D86+ DC (% / MFI) in different cohorts (NC vs CHB vs HBV-ACLF) have been provided, only t-SNE plots; this is important re. HVEM-BTLA axis. Is HVEM expression altered in other APCs e.g., monocytes?

Authors' response: Thank you for raising this important point. We have presented HVEM expression (MFI) on CD80/D86+ DC and monocytes in different cohorts (NC vs CHB vs HBV-ACLF) in Supplemental Fig. 2e. The expression of HVEM on CD80/D86+ DC and monocytes was increased in patients with HBV-ACLF compared with that in NC and CHB patients (lines 149–151).

Supplemental Fig. 2 Expression of BTLA in ACLF patients of various etiologies, as well as BTLA and HVEM expression on T cells, NK cells, and dendritic cells (DC). (e) MFI of HVEM expression on CD4/CD8+ T cells but not on NK cells decreased in patients with HBV-

ACLF ($n = 35$) compared with that in NC ($n = 20$); HVEM levels on CD80/D86⁺ DC and monocytes were increased in patients with HBV-ACLF compared with those in NC and CHB patients ($n = 27$).

3. Fig. 3D: BTLA expression (%) in CD4⁺ T cells do not seem majorly reduced; the significance shown here is probably due to one sample only. What about BTLA expression at MFI level?

Authors' response: Thank you for correctly pointing this out. After treatment, there was a more significant decrease in BTLA expression (MFI) on CD4⁺ T cells compared to its frequency of expression (lines 234–237).

Fig. 3 Elevated BTLA expression on CD4⁺ T cells was induced by circulating inflammation cytokines in HBV-ACLF. (c) Comprehensive treatment ($n = 9$) decreased the levels of IL-6 and TNF- α , as well as BTLA expression on CD4⁺ T cells.

4. Fig. 3: The authors demonstrate that treatment of NC PBMCs with HBV-ACLF (vs CHB or NC) plasma induces BTLA upregulation in CD4⁺ T cells. However, this experimental set up doesn't show plasma's direct effects on CD4⁺ T cells (e.g., it could be immune cell-mediated effects on CD4⁺); this could be further addressed by culturing NC-derived CD4⁺ T cells with plasma. Given the CD4⁺ BTLA's correlation with plasma IL-6 and TNF- α , and by performing cell culture with rhIL-6 or rhTNF- α , the authors propose that these cytokines mechanistically induce BTLA expression in CD4⁺ cells in HBV-ACLF. To support this stronger, the authors should explore in vitro the blockade of IL-6 or TNF- α in plasma (e.g., anti-IL-6 or anti-TNF- α mAb), using both PBMCs and isolated CD4⁺ T cells from NC. Also, these two cytokines drive Th1; one would wonder if BTLA is Th1 related? e.g., same trends in IL-1 β treatment (but plateaued).

Authors' response: We thank you for your insightful comment. To provide a mechanistic explanation for the upregulation of BTLA expression in patients with HBV-ACLF, we further investigated the effect of HBV-ACLF plasma on BTLA expression in CD4⁺ T cells from NC patients. Exposure of purified CD4⁺ T cells from NC to HBV-ACLF plasma resulted in a higher BTLA expression in CD4⁺ T cells compared with that in purified CD4⁺ T cells exposed to NC plasma, while anti-IL-6 or (plus) anti-TNF- α (not anti-IL-1 β) reduced the HBV-ACLF plasma-induced BTLA upregulation in CD4⁺ T cells (Fig. 3d, lines 239–242).

Elevated BTLA expression on CD4⁺ T cells was induced by circulating inflammation cytokines in HBV-ACLF. (d) Exposure of purified CD4⁺ T cells from NC to HBV-ACLF plasma ($n = 13$) resulted in a higher BTLA expression on CD4⁺ T cells compared with exposure of PBMC to NC plasma, while HBV-ACLF plasma combined with anti-IL-6 and/or anti-TNF- α resulted in decreased BTLA expression on CD4⁺ T cells.

5. Fig. 4: In addition to activation markers (e.g., CD25, CD69) or cytokines screened, do CD4⁺ T cells differentially express immune checkpoints (e.g., PD-1, CTLA-4, TIGIT, Tim3) following BTLA agonism? Also, in (Fig. 4C) the CFSE analysis needs clarification; how is it calculated, what is “CFSE %”

Authors' response: Thank you for raising this important issue. We found that crosslinking of BTLA did not result in changes in the expression of other immune checkpoints on CD4⁺ T cells, such as programmed cell death protein-1 (PD-1), cytotoxic T-lymphocyte antigen 4 (CTLA-4), T-cell immunoglobulin, mucin-domain-containing-3 (TIM-3), or T-cell immunoglobulin and immunoreceptor tyrosine-based inhibitory motif domain (TIGIT; Supplemental Fig. 7c; lines 269–272).

Supplemental Fig. 7 Time- and dose-dependent inhibition of CD4⁺ T-cell activation by anti-BTLA. (c) Crosslinking of BTLA did not result in changes in the expression of programmed cell death protein-1 (PD-1), cytotoxic T-lymphocyte antigen 4 (CTLA-4), T-cell immunoglobulin and mucin-domain-containing-3 (TIM-3), or T-cell immunoglobulin and ITIM domain (TIGIT).

Carboxyfluorescein succinimidyl ester (CFSE) is a versatile tool for the fluorescent intracellular labeling of live cells. T cell proliferation was detected by flow cytometry with CFSE-labeled T cells. The discrete peaks in these histograms represent successive generations of life. The unstimulated parent generation is indicated in blue, and discrete peaks on the left represent the percentage of proliferation (CFSE%; lines 290–291).

Figure. Cell proliferation was followed for 7 generations using the CellTrace™ CFSE reagents.

6. Fig. 5D (lines 256-257): *BTLA shRNA-treated CD4⁺T cells tended to show higher apoptosis levels than the control shRNA-treated CD4⁺T cells; data show the opposite, needs correction.*

Authors' response: Thank you for pointing out this error; we have revised it accordingly (lines 294–295).

Fig. 5 BTLA knockdown increased activation, production of cytokines, and proliferation of CD4⁺ T cells but abrogated apoptosis. (c) Contour plots (up) and line graphs (down) display frequencies of Annexin V⁺CD4⁺ T cells in the presence of control shRNA and BTLA shRNA.

7. Fig. 5E: *CD69 expression is at 60% on culture Day 3; it should drop after 12 hours, this needs explanation; the authors could provide some representative flow cytometry plots for the stains assessed e.g., CD69 (same for anti-mouse CD69).*

Authors' response: As shown in Supplemental Fig. 7a, CD69 expression on CD4⁺ T cells reached its peak (about 70–80%) at the 12th hour of culture and then gradually decreased to 60–70% on day 1 and reached 60% on day 3, which was exactly in line with the expression of CD69 on day 1 in Fig. 5e (about 60%).

We also show representative flow cytometry plots for the expression of CD38 on CD4⁺ T cells from peripheral blood of NC (Fig. 5e; lines 293–294) and ConA-induced ACLF mouse models (Supplementary Fig. 11c; lines 327–331).

Supplemental Fig. 7 Time- and dose-dependent inhibition of CD4⁺ T-cell activation by anti-BTLA. (a) Crosslinking of BTLA showed the strongest ability to suppress CD4⁺ T cell activation upon 1 day of anti-BTLA stimulation ($n = 2$).

Fig. 5 BTLA knockdown increased activation, production of cytokines, and proliferation of CD4⁺ T cells but abrogated apoptosis. Expression of CD25, CD38, and CD69 on CD4⁺ T cells activated by anti-CD3/CD28 for 1 day in the presence of control shRNA and BTLA shRNA. All PBMC were isolated from NC ($n = 6$). * $P < .05$.

Supplemental Fig. 11 BTLA expression significantly increased on circulating CD4⁺/CD8⁺ T cells in ACLF model induced by ConA. (c) Contour plots showing that the percentages of activation indices (CD25, CD38, and CD69) were higher in BTLA^{-/-} mice than in WT mice following ConA injection on day 14.

8. Fig. 7: The authors used a ConA (8 mg/kg) murine model to mimic ACLF and compare WT vs BTLA KO animals. However, the model is poorly characterized here: 1) histological stains: a) only representative H&E stains shown; what about quantitative data? the areas of liver damage/necrosis (this can be quantified as % of area) or the inflammatory cell infiltrate (histopathological score), b) PicroSiriusRed (PSR) stains re. liver fibrosis have not been included; this could be measured as area of collagen (%). How do these differ in WT vs BTLA mice? Other indices of liver injury should be measured e.g., serum ALT, AST levels.

Authors' response: We thank you for this insightful suggestion. In addition to H&E staining, we also used reticular fiber staining and Masson staining to label liver fibers. Meanwhile, we also determined and compared liver damage indexes (ALT, AST, and TBil), liver inflammation (or necrosis), and fibrosis scores in WT and BTLA^{-/-} mice (lines 323–325, Supplemental Fig. 10a–d) as follows:

Supplemental Fig. 10 Characterization of a mouse model of ACLF induced by Concanavalin A (ConA). (a, b) Reticular fiber (left) and Masson staining (right) of liver pathology at baseline, 8 days, and 14 days in WT and BTLA^{-/-} C57BL/6 mice. (c, d) Serum ALT, AST, and TBil levels, inflammation, and fibrosis scores were measured at 8 and 14 days post-ConA injection. (e) Cytokine (TNF- α , IL-6, and IFN- γ) levels were slightly increased, while IL-10 levels were slightly decreased in the plasma of BTLA^{-/-} mice compared to those in WT mice at day 14.

9. Figure 7 2) Re. flow cytometry: representative gating strategies to stain and identify mouse liver CD4⁺ T cells and measure BTLA are not included; similarly, to what was explored in human liver, is BTLA differentially expressed in mouse liver CD8⁺ or NK cells following ConA-induced injury? Representative flow cytometry stains of CD25, CD38, CD69 and cytokines should be included (probably in suppl. material).

Authors' response: Thank you for pointing this out. The expression of BTLA on CD8⁺ T cells was also significantly increased in WT mice following ConA injection (Supplemental Fig 11a, b; line 327). The representative gating strategies of BTLA expression on T cells and activation indices (CD25, CD38, and CD69) and cytokines (IFN- γ and TNF- α) are included in Supplemental Fig. 11c, d (lines 328–335).

The representative gating strategies of BTLA expression on intrahepatic CD4⁺ T cells in NC, CHB, and HBV-ACLF patients (Supplemental Fig. 2c; lines 141–142).

Supplemental Fig. 11 BTLA expression significantly increased on circulating CD4⁺/CD8⁺ T cells in ACLF model induced by ConA. (a) Flow cytometry diagram of BTLA-expressing CD4⁺/CD8⁺ T cells from peripheral blood of WT mice. (b) Expression of BTLA on CD8⁺ T cells was significantly increased on days 8 and 14, compared to the baseline, in WT mice following ConA injection. (c, d) Contour plots showing that the percentages of activation indices (CD25, CD38, and CD69) and cytokines (IFN- γ and TNF- α) were higher in BTLA^{-/-} mice than in WT

mice following ConA injection on day 14.

Supplemental Fig. 2 Expression of BTLA in ACLF patients of various etiologies, as well as BTLA and HVEM expression on T cells, NK cells, and dendritic cells (DC). (c) Flow cytometry diagram of BTLA expression on intrahepatic CD4⁺ T cells in NC, CHB, and HBV-ACLF patients.

10. Figure 7 3) Given that following CLP, the mortality rate and bacDNA levels were decreased in BTLA KO mice (compared to WT) the therapeutic *in vivo* blockade of BTLA (e.g., administration of anti-mouse BTLA mAb following induction ConA-induced injury, or prior to liver injury) should be also investigated to assess its clinical use potential (translation into future therapy).

- The CLP sepsis after ConA-induced liver injury should be studied in more detail (it could be Fig. 8 on its own). In addition to mortality and bacDNA, the authors should also examine tissue histology (e.g., liver, kidney, spleen, lungs), serum cytokines (pro- or anti-inflammatory) and indices of liver injury/damage (H&E/PSR stains, serum ALT/AST).

Authors' response: We thank you for your insightful comments and suggestions.

To address concerns raised by certain experts regarding the recognition of the ConA-induced model as an ACLF model, we opted to adopt an established method from the literature⁵ to construct an ACLF mouse model. This method involves inducing chronic liver injury through eight weeks of continuous CCl₄ injection, followed by a double dose of CCl₄ to provoke acute liver injury. This approach was chosen to provide a robust platform for investigating the impact of BTLA knockout or blocking on reducing mortality and infection (Fig. 8a). The tissue histology (e.g., liver), serum cytokines (pro/anti-inflammatory), and indices of liver injury/damage (H&E/Masson/Reticular fiber staining, serum ALT/AST) indicated the successful induction of an ACLF model (Fig. 8d and Supplemental Fig. 12a–e). We did find that injection of anti-mouse BTLA Ab reduced mortality and bacterial load in ACLF mice before injection of *K.P.* (Fig. 8g, h). Therefore, injection of anti-mouse BTLA Ab is a potential treatment for ACLF (lines 344-368).

Due to the relatively short survival time of mice following CLP, it was imperative to minimize blood extraction, as excessive sampling could accelerate mortality and potentially interfere with our experimental observations. Therefore, we collected only a small volume of blood for bacterial DNA testing. However, it is worth noting that, prior to CLP induction, we collected blood samples and liver tissues at various time points to analyze changes in biochemical

markers (ALT/AST), cytokine levels (IL-6, IL-10, IFN- γ , and TNF- α), as well as liver pathology through H&E, Masson, and Reticular fibers staining (lines 320-325, Supplemental Fig. 10a-e).

Fig. 8 BTLA contributed to immune exhaustion and increased mortality and infection rates in the ACLF model induced by CCl₄. (a) Schematic diagram of ACLF induced by continuous injection of CCl₄ (0.2 ml/kg, twice a week) for 8 weeks and subsequent injection of a double dose of CCl₄ (0.4 ml/kg). (b) Hematoxylin-eosin diagram of liver pathology at baseline, 8 weeks, and 8 week + 3 days in WT and BTLA^{-/-} C57BL/6 mice. (c) Cytokine (TNF- α , IL-2, and IFN- γ) production in CD4⁺ T cells was higher in BTLA^{-/-} mice than in WT mice at 6 and 8 weeks ($n = 10$). (d) Cytokine (TNF- α , IL-2, and IFN- γ) levels were markedly increased in the plasma of BTLA^{-/-} mice compared to those in WT mice after a double dose of CCl₄ injection. (e, g) Survival rate and K.P. load in peripheral blood of both BTLA^{-/-} and WT mice injected with anti-BTLA antibodies were higher than those in the WT and anti-IgG injection groups.

Supplemental Fig. 10 Characterization of a mouse model of ACLF induced by Concanavalin A (ConA). (a, b) Reticular fiber (left) and Masson staining (right) of liver pathology at baseline, 8 days, and 14 days in WT and BTLA^{-/-} C57BL/6 mice. (c, d) Serum ALT, AST, and TBil levels, inflammation, and fibrosis scores were measured at 8 and 14 days post-ConA injection. (e) Cytokine (TNF-α, IL-6, and IFN-γ) levels were slightly increased, while IL-10 levels were slightly decreased in the plasma of BTLA^{-/-} mice compared to those in WT mice at day 14.

11. Importantly, the ConA (8 mg/kg) model is different, modified to mimic ACLF in mice, to the one described in text (ref. 24: ConA in rats), therefore it's not well-established. A well-characterized murine ACLF model in the field is the CCl₄-induced liver fibrosis (6-10w) model in combination with *i.p.* bacterial infection (e.g., *E. coli*) or *i.p.* LPS (PMID: 31987990 and PMID: 31786256). This model hasn't been examined in this manuscript.

Authors' response: Thank you for your insightful comment. We adopted the method described in

the literature to construct a mouse model of ACLF⁵, that is, continuous injection of CCl₄ for 8 weeks to form chronic liver injury, and then 3 days later, applied a double dose of CCl₄ to induce acute liver injury (lines 344–368), as shown in Fig. 8a–h and Supplemental Fig. 12a–e.

Supplemental Fig. 12 Characterization of a mouse model of ACLF induced by CCl₄. (a, b) Reticular fiber (left) and Masson staining (right) of liver pathology at baseline, 8 weeks, and 8 weeks + 3 days in WT and BTLA^{-/-} C57BL/6 mice. (c, d) Serum ALT, AST, and TBil levels, inflammation, and fibrosis scores were measured at 8 weeks post-chronic CCl₄ injection and 3 days post-acute CCl₄ injection. (e) Level of IL-10 was significant at 6 weeks post-chronic CCl₄ injection.

Reviewer #2 (expertise in liver Immunology):

Comments

1. CD4 T cell responses play an important role in HBV infection. Moreover, ACLF is a highly complex systemic condition. In this manuscript it generally remains unsolved whether the phenomenon of BTLA expression and T cell exhaustion described was due to HBV, HBV induced cirrhosis, ACLF or superimposed bacterial infection? Does it also occur in ACLF of other etiology? Or in intensive care patients with other infections? What of these components exactly leads to BTLA expression presumably mediated by TNF and IL6?

Authors' response: We thank you for these insightful questions. Our study shows that compared to NC and CHB patients, patients with HBV-ACLF had higher expression of BTLA on CD4⁺ T cells, and patients with HBV-ACLF with co-infection had higher expression of BTLA. Therefore, it can be said that ACLF, or superimposed bacterial infection, induces BTLA expression and immune exhaustion. Because many patients with HBV-ACLF also had cirrhosis, HBV-induced cirrhosis could not be ruled out. To further clarify whether other etiologies besides HBV contribute to the same phenomenon, we analyzed the expression of BTLA on CD4/CD8⁺ T cells in NC (*n* = 38), CHB (*n* = 93), HBV-ACLF (*n* = 35), alcohol-induced ACLF and cirrhosis (*n* = 14), and primary biliary cholangitis (PBC) patients (*n* = 4), and found that BTLA was also highly expressed on CD4⁺ T cells in patients with alcohol-related ACLF or cirrhosis. Therefore, it can be considered that CD4⁺ T cells have high expression of BTLA in ACLF brought on by various causes (lines 129–132).

There may be many reasons for increased BTLA expression. In this study, pro-inflammatory cytokines such as TNF- α and IL-6 were demonstrated to induce BTLA expression through in vivo and in vitro experiments (lines 239–250).

Supplemental Fig. 2 Expression of BTLA in ACLF patients of various etiologies, as well as BTLA and HVEM expression on T cells, NK cells, and dendritic cells (DC). (a, b) Expression of BTLA on CD4/CD8⁺ T cells in NC (*n* = 38), CHB (*n* = 93), HBV-ACLF (*n* = 35), alcohol-induced ACLF and cirrhosis (*n* = 14), and primary biliary cholangitis (PBC) patients (*n* = 4).

2. In Figure 1 E-H (Results page 7/8, line 133 onwards) it remains unclear whether CD4⁺ T cells here are derived from the blood or the liver.

Authors' response: Thank you for raising this important point. The CD4⁺ T cells in Fig. 1 e–h are derived from the peripheral blood (line 157).

3. Figure 1 H/ Supplemental 5E: what is the definition of “good” and “poor” prognosis?

Authors' response: Thank you for raising this important point. Patients with ACLF were divided into two groups based on their clinical outcome: good and poor prognosis. The good prognosis

group included patients whose liver function and blood coagulation gradually recovered and who were still alive after 6 months. The poor prognosis group included patients whose liver function deteriorated progressively and who died or received liver transplantation with poor response to comprehensive treatment within 6 months (lines 177–183).

4. Overall it appears that the BTLA⁺ and BTLA⁻CD4⁺ T cells are very similar, what do the authors think is the biological significance? Figure 2B/C: Are the data obtained from trace RNA-seq and flow cytometry contradictory? At least differences in CD7, CD127, CD25 seem to be detected only in flow cytometry. Figure 2E does not seem to reveal significant differences in the studied population of ACLF.

Authors' response: The previous results showed that BTLA expression was elevated on CD4⁺ T cells in patients with HBV-ACLF, and was correlated with disease severity and prognosis. Therefore, we explained the relationship between BTLA and disease progression by comparing the functional phenotypes (such as activation, proliferation, and survival) of BTLA⁺CD4⁺ T and BTLA⁻CD4⁺ T cells.

First, we detected two pairs of samples using trace RNA-seq to identify which genes had differential expression and then verified them on NC (n = 38), CHB (n = 94), and HBV-ACLF (n = 35) samples using flow cytometry. Trace RNA-seq results showed that the expression of CD7, CD127, and CD25 in BTLA⁺CD4⁺ T cells was slightly higher than that in BTLA⁻CD4⁺ T cells, but there was no statistical difference. Through the verification column, it was obvious that the expression of CD7, CD127, and CD25 increased on BTLA⁺CD4⁺ T cells. Therefore, the data obtained from trace RNA-seq and flow cytometry are not contradictory.

In NC, the proportion of Tem and TEM-RA subtypes (representing differentiated subsets of CD4⁺ T cells) in BTLA⁺CD4⁺ T cells was higher than that in BTLA⁻CD4⁺ T cells, while there were no significant differences in the proportion of Tem and TEM-RA between BTLA⁺CD4⁺ T and BTLA⁻CD4⁺ T cells in patients with CHB and HBV-ACLF. This indicates that the differentiation degree of BTLA⁺CD4⁺ T cells in patients with HBV-ACLF was decreased compared with that of BTLA⁻CD4⁺ T cells.

5. The upregulation of diverse cytokines in ACLF (Figure 3B) has been repeatedly described before.

Authors' response: Thank you for pointing this out. Indeed, many studies have reported cytokine storms in patients with HBV-ACLF, but the results may be biased due to sample size, detection method, and/or retention time. Therefore, this study used Luminex technology to detect 17 anti-inflammatory/pro-inflammatory cytokines in order to obtain more objective data.

6. Concanavalin A (ConA) is a model for acute immune-mediated hepatitis such as viral or autoimmune hepatitis. Apart from its initial description in the cited reference (Ref 24), it has not become a recognized ACLF model (Ref doi: 10.14218/JCTH.2022.00086). Also blocking BTLA has been shown beneficial in another mouse model of acute hepatitis (MHV-3, Ref 14). It is unclear why the authors did not choose a more established animal model to simulate ACLF?

Authors' response: Thank you for raising this important point. We adopted the method described in the literature to construct a mouse model of ACLF, that is, continuous injection of CCL4 for 8 weeks to form chronic liver injury, and then use a double dose of CCl₄ to induce acute liver injury

to simulate ACLF 3 days later (lines 344–368, Fig. 8a–h and Supplemental Fig. 12a–e).

Fig. 8 BTLA contributed to immune exhaustion and increased mortality and infection rates in the ACLF model induced by CCl₄. (a) Schematic diagram of ACLF induced by continuous injection of CCl₄ (0.2 ml/kg, twice a week) for 8 weeks and subsequent injection of a double dose of CCl₄ (0.4 ml/kg). (b) Hematoxylin-eosin diagram of liver pathology at baseline, 8 weeks, and 8 week + 3 days in WT and BTLA^{-/-} C57BL/6 mice. (c) Cytokine (TNF-α, IL-2, and IFN-γ) production in CD4⁺ T cells was higher in BTLA^{-/-} mice than in WT mice at 6 and 8 weeks (n = 10). (d) Cytokine (TNF-α, IL-2, and IFN-γ) levels were markedly increased in the plasma of BTLA^{-/-} mice compared to those in WT mice after a double dose of CCl₄ injection. (e, g) v(f, h) Survival

rate and K.P. load in peripheral blood of both $BTLA^{-/-}$ and WT mice injected with anti-BTLA antibodies were higher than those in the WT and anti-IgG injection groups.

Supplemental Fig. 12 Characterization of a mouse model of ACLF induced by CCl_4 . (a, b) Reticular fiber (left) and Masson staining (right) of liver pathology at baseline, 8 weeks, and 8 weeks + 3 days in WT and $BTLA^{-/-}$ C57BL/6 mice. (c, d) Serum ALT, AST, and TBil levels, inflammation, and fibrosis scores were measured at 8 weeks post-chronic CCl_4 injection and 3 days post-acute CCl_4 injection. (e) Level of IL-10 was significant at 6 weeks post-chronic CCl_4 injection.

7. Discussion line 318: the Ref Chen et al seems relevant but is missing in the list of publications.

Authors' response: Thank you for pointing this out. We have added this reference to the list (line 602-604), as follows:

18. Xu H, Cao D, Guo G, Ruan Z, Wu Y, Chen Y. The intrahepatic expression and distribution of BTLA and its ligand HVEM in patients with HBV-related acute-on-chronic liver failure. *Diagnostic Pathology* 7, 142 (2012).

8. Methods: 104 CHB patients and 101 severe liver injury – 71 due to HBV – were the other liver injury patients also analyzed? Please state this accurately. Were the 9 controls age matched?

Authors' response: In total, we included 104 patients with CHB and 101 patients with severe liver injury (TBil \geq 5 mg/dL and INR \geq 1.5), all of which were HBV-positive. Among the 101 cases of severe liver injury, 71 patients met the diagnostic criteria of ACLF and were included in the analysis, while the other 30 patients did not meet the criteria and were therefore excluded (lines 491–496).

We believe that your question here is regarding the age matching between the 90 healthy controls and 104 CHB and 71 HBV-ACLF patients. Patients with HBV-ACLF (average age: 45 years; range 35–52 years) were older than those in the CHB (average age: 31 years; range 26–41 years) 31 (26–41) and HC groups (average age: 30 years; range 26–44 years) 30 (26–44). Because HBV-ACLF typically represents an advanced stage of liver disease that often develops from CHB and may even progress to liver cirrhosis, patients with HBV-ACLF tend to be of older age.

Minor comments:

1. *The abbreviation ILC is reserved for “innate lymphoid cells” and therefore should not be used to abbreviate “intrahepatic lymphocytes cells”. Also in Figure 1C these cells are then termed “liver infiltrating lymphocytes”. Consistency with terms is missing.*

Authors' response: Thank you for your careful review of our manuscript; we have changed it to “liver-infiltrating lymphocytes,” abbreviated as LIL (lines 140–141).

2. *Overall the language needs review by a native speaker.*

Authors' response: Thank you for your advice. The manuscript text has been reviewed by a native English speaker.

3. *Page 6, line 113 “interestingly, ...”*

Authors' response: Thank you for pointing out this error; it has been corrected to “Notably” (line 134).

4. *page 8, line 147 – what are “coinfection patients” HBV/HDV coinfecting have been excluded?! Or bacterial infection superimposed on HBV-ACLF? Please specify.*

Authors' response: Patients who had HBV co-infection with HAV, HCV, or HDV were excluded. Patients with superimposed bacterial infection of HBV-ACLF were not excluded (lines 493–494).

5. *Page 10, line 203: typo “the-6”; wording line 207.*

Authors' response: Thank you for pointing out this error; it has been corrected (line 229).

6. *Suppl Fig8 G,H,I: anti-BTLA should always be displayed on the same side.*

Authors' response: Thank you for your careful review of our manuscript; we have revised it.

Supplemental Fig. 9 Anti-BTLA crosslinking increased gene expression changes in NC ($n = 3$), CHB ($n = 3$), and HBV-ACLF patients ($n = 3$). Correlation, volcano plot, and heatmap of gene expression of PBMC with or without anti-BTLA crosslinking from NC (a, d, g), CHB (b, e, h), and HBV-ACLF patients (c, f, i) are shown.

Reviewer #3 (expertise in hepatology and liver failure):

Major aspects:

1. *ACLF classification: ACLF scoring of included patients was performed using the COSSH-ACLF score. The classification is different to the EF CLIF classification regarding two main aspects:*

1) *In contrast to the EASL-CLIF ACLF score, the COSSH score includes patients with HBV-related ACLF with and without liver cirrhosis. The authors explicitly state here that patients were included “regardless of presence of cirrhosis” (l. 408). That means, that within the patients’ “HBV-ACLF” cohort might also include patients with acute liver failure. Therefore, the described results for the role of BTLA are not specific for ACLF.*

2) *Failure of each organ system is assessed using the CLIF-C Organ Failure Score in both cases, however and in contrast to the EASL-CLIF score, according to the COSSH-ACLF patients with single liver failure and either $INR > 1.5$ OR creatinine ranging from 1.5-1.9 are classified as ACLF grade 1 but would not meet the criteria for ACLF I° according to the EF-CLIF classification. The authors should discuss what impact the classification using the COSSH-ACLF score has in comparison to other internationally recognized score (EASL-CLIF, NACSELD) regarding the results of their work and whether the functional and prognostic relevance of BTLA is valid also if the included patients are scored by EF-CLIF ACLF score.*

Authors' response:

1) We thank you for your insightful questions and comments. According to China's Guidelines for the Diagnosis and Treatment of Liver Failure, acute liver failure refers to acute onset within 2 weeks and generally has no basis for chronic liver disease. The patients included in the COSSH criteria have chronic HBV, and thus their condition can be considered liver failure on the basis of chronic liver disease. Therefore, COSSH criteria excludes patients with acute liver failure. However, in addition to HBV etiology, alcohol etiology was also included in this study, and the results showed that BTLA was highly expressed on CD4⁺ T cells in both patients with HBV and alcohol-related ACLF, so the role of BTLA was not limited to HBV etiology (Supplemental Fig. 1a; lines 124–132).

2) The CLIF-OFs ACLF classification grade 1 is as follows: (1) patients with single kidney failure; (2) patients with single failure of the liver, coagulation, circulation, or respiration who had a serum creatinine level ranging from 1.5 to 1.9 mg/dL and/or mild to moderate hepatic encephalopathy; and (3) patients with single cerebral failure who had a serum creatinine level ranging from 1.5 to 1.9 mg/dL. In comparison, the COSSH-ACLF classification grade 1 is as follows: (1) patients with kidney failure alone; (2) patients with single liver failure with an $INR \geq 1.5$ and/or kidney dysfunction and/or HE grade I or II; (3) patients with a single type of organ failure of the coagulation, circulatory, or respiratory systems and/or kidney dysfunction and/or HE grade I or II; and (4) patients with cerebral failure alone plus kidney dysfunction. Therefore, compared with the CLIF-OFs diagnostic criteria, the CHSSH diagnostic criteria seem to have a wider range, mainly reflected in the fact that kidney injury is not limited to creatinine in the 1.5 to 1.9 range.

In the Asia-Pacific region (particularly in China), HBV infection is the main etiology of ACLF, and research has suggested that a significant proportion of patients with ACLF are non-cirrhotic or do not have extrahepatic organ failures upon admission^{6, 7}. Therefore, compared to the EASL-CLIF and NACSELD diagnostic criteria, COSSH-ACLF is more suitable with this study.

However, in order to clarify the impact of different diagnostic criteria for ACLF, we included an ACLF study population based on EASL-CLIF⁸ and NACSELD⁹ diagnostic criteria. Among 101 patients with severe liver injury, 78 met the criteria of EASL-CLIF and 18 met the criteria of NACSELD. As shown in Supplemental Fig. 1a and b, patients with HBV-ACLF who met the diagnostic criteria of NACSELD or EASL-CLIF had significantly increased expression of BTLA on peripheral blood CD4⁺ T cells compared to NC patients and those with CHB, while there was no significant difference in the BTLA expression of CD8⁺ T cells. The MFI of BTLA expression on CD4⁺ T cells was also positively correlated with the severity of the disease (Child-pugh, MELD scores, CLIF-SOFA, CLIF-C ACLFs, and COSSH-ACLFs). The MFI of CD4⁺BTLA⁺ T cells was associated with the complication, prognosis, and HBeAg status of HBV-ACLF. More importantly, the patients with a good prognosis of HBV-ACLF had a significantly decreased MFI of CD4⁺BTLA⁺ T cells after 4 weeks compared with that before treatment. These results conform to the COSSH diagnostic criteria of ACLF, indicating that the function and predictive value of BTLA are also applicable to patients with COSSH-ACLF (lines 124–129).

Supplemental Fig. 1 BTLA expression was significantly increased on circulation CD4⁺ T cells in HBV-ACLF patients and was positively correlated with prognosis and infectious complications. (a, b) Patients with HBV-ACLF who met the diagnostic criteria of NACSELD (a, NC: *n* = 90, CHB: *n* = 104, HBV-ACLF: *n* = 18) or EASL-CLIF (b, NC: *n* = 90, CHB: *n* = 104, HBV-ACLF: *n* = 78) had significantly increased expression of BTLA on peripheral blood CD4⁺ T cells compared to NC and CHB patients, while there was no significant difference in the BTLA

expression of CD8⁺ T cells. (c) The MFI of BTLA expression on CD4⁺ T cells was positively correlated with the severity of the disease (Child-pugh, MELD scores, CLIF-SOFA, CLIF-C ACLFs, and COSSH-ACLFs, $n = 78$). (d) Relationship between the MFI of CD4⁺BTLA⁺ T cells and complications, prognosis, and HBeAg status. (e) Changes in the MFI of CD4⁺BTLA⁺ T cells in the progression of HBV-ACLF (EASL-CLIF score, $n = 78$). (f) The patients with a good prognosis of HBV-ACLF ($n = 14$) had a significantly decreased MFI of CD4⁺BTLA⁺ T cells after 4 weeks compared with that before treatment.

2. *Generality of results: The authors describe a prognostic relevance of BTLA in HBV-ACLF. However, as the main findings of the study relate to CD4 T cells, the findings are not specific for HBV-related liver diseases or viral immune defense. Therefore, the authors should investigate whether the upregulation of BTLA on CD4⁺ cells is also present within other ACLF etiologies to demonstrate the generality of the biological role of BTLA in ACLF. Therefore, a second cohort of patients with another etiology of cirrhosis, i.e. alcoholic cirrhosis, have to be included in the analysis and findings need to be reproduced here.*

Authors' response: We thank you for your insightful comments. To further confirm that the upregulation of BTLA on CD4⁺ T cells can also be found in patients with ACLF with other etiologies, we analyzed the expression of BTLA on T cells in 38 healthy controls, 93 patients with CHB, 35 patients with HBV-ACLF, 14 patients with alcohol-induced ACLF and cirrhosis, and 4 patients with primary biliary cholangitis (PBC). The results showed that in addition to the high expression of BTLA on CD4⁺ T cells of patients with HBV-ACLF, patients with alcohol-induced ACLF and cirrhosis also showed high expression of BTLA on CD4⁺ T cells, while there was no high expression of BTLA in patients with PBC (which may be related to there being only 4 cases of PBC). There was no difference in the expression of BTLA on CD8⁺ T cells in NC, CHB, HBV-ACLF, alcohol-induced ACLF, cirrhosis, or PBC patients (Supplemental Fig. 2a, b; lines 129–132).

Supplemental Fig. 2 Expression of BTLA in ACLF patients of various etiologies, as well as BTLA and HVEM expression on T cells, NK cells, and dendritic cells (DC). (a, b) Expression of BTLA on CD4/CD8⁺ T cells in NC ($n = 38$), CHB ($n = 93$), HBV-ACLF ($n = 35$), alcohol-induced ACLF and cirrhosis ($n = 14$), and primary biliary cholangitis (PBC) patients ($n = 4$).

3. *Methodology: Please also provide MFI data for HVEM expression on CD80/86⁺ DCs as complementation for subfigure 1D.*

Authors' response: Thank you for pointing this out. We have provided the MFI data for HVEM expression on CD80/86⁺ DCs as a complement to Supplemental Fig. 1d.

Supplemental Fig. 2 Expression of BTLA in ACLF patients of various etiologies, as well as BTLA and HVEM expression on T cells, NK cells, and dendritic cells (DC). HVEM levels on CD80/D86⁺ DC and monocytes were increased in patients with HBV-ACLF compared with those in NC and CHB patients ($n = 27$).

4. CD25: Figure 2C shows an upregulation of CD25 in MFI on CD4⁺ BTLA⁺ T cells in ACLF patients. The authors argue that the BTLA expression is accompanied by an upregulation of CD25 as a marker for T cell activation. In contrast, in the *in vivo* model an upregulation of CD25 is seen on CD4⁺ T cells in BTLA *ko* mice. How do these findings fit together? This should be further discussed.

Authors' response: Thank you for raising these important points. As shown in Fig. 8d, the expression of CD25 in the BTLA^{-/-} ACLF mouse model was significantly higher than that in the WT ACLF mouse model. The main reason was the presence of the BTLA ligand HVEM in WT mice, which inhibited the expression of CD25 when combined with BTLA, while CD25 expression in BTLA^{-/-} mice was not inhibited. In Fig. 2c, CD4⁺BTLA⁺ T cells from patients with HBV-ACLF expressed higher levels of CD25 than CD4⁺BTLA⁻ T cells, which seems to be contrary to the results shown in Fig. 7d. However, this was an *in vitro* experiment that directly compared the difference in CD25 expression on CD4⁺BTLA⁺ T and CD4⁺BTLA⁻ T cells and did not use functional antibodies (anti-BTLA antibodies) to cross-link BTLA or activate the BTLA signaling pathway to inhibit CD25 expression. Therefore, it appears that CD4⁺BTLA⁺ T cells express higher levels of CD25 than CD4⁺BTLA⁻ T cells. In the case of CD4⁺BTLA⁺ T and CD4⁺BTLA⁻ T cells being sorted *in vitro*, anti-BTLA antibody crosslinked or non-crosslinked CD4⁺BTLA⁺ T cells were used for 24 h; the results showed that the crosslinked CD4⁺BTLA⁺ T cells expressed lower levels of CD25 than the non-crosslinked CD4⁺BTLA⁺ T cells, and the levels were even lower than those expressed by CD4⁺BTLA⁻ T cells (as shown in the following figure), which are consistent with the results shown in Fig. 7d. Therefore, the difference between Fig. 2c and Fig. 7d is mainly attributed to the difference between *in vitro* and *in vivo* experiments (whether BTLA is cross-linked or activated to initiate inhibitory pathways).

Figure. CD4⁺BTLA⁺ T and CD4⁺BTLA⁻T cells were sorted and cultured with anti-CD3 /CD8 antibodies with or without anti-BTLA for 24 h. The expression of CD25 in BTLA⁺ T cells was higher than that in BTLA⁻ T cells. In crosslinked BTLA⁺ T cells, CD25 expression was lower than that in non-crosslinked BTLA⁺ T cells and BTLA⁻T cells.

Minor aspects:

1. *PBMC sampling: What was the time point of blood draw for PBMC isolation? Was this hospital admission and was there a maximum of hospital admission duration (i.e. within first 72h of admission) until when the PBMC had to be collected? If not sampling error could occur due to therapeutic measures and already resolved ACLF, especially in low grade ACLF patients. The authors should comment on this.*

Authors' response: Blood samples of all patients were collected and PBMC were isolated within 24 h post-admission (line 478–479).

2. *Line 152: do you mean survivors and non-survivors by “good prognosis” and “poor prognosis”? In your previous publication you are referring to (#22) the groups are called survivors and non-survivors. Please stick to that labeling if these are the same groups, if not, please state in the text in more detail how these groups are characterized and how these groups then refer to your previous publication*

Authors' response: Thank you for pointing this out. Patients with ACLF were divided into two groups by their final clinical outcome: good and poor prognosis. The good prognosis group included patients whose liver function and blood coagulation gradually recovered and who were still alive after 6 months. The poor prognosis group included patients whose liver function deteriorated progressively and who died or received liver transplantation with poor response to comprehensive treatment within 6 months (lines 177–183).

3. *Please add the exact number of ACLF patients (71) to the results part.*

Authors' response: Among 101 patients with severe liver injury (TBil \geq 5 mg/dL and INR \geq 1.5), 71 patients with HBV-ACLF were enrolled in this study (lines 491–493).

4. *Please correct “interesting” in line 113 to “interestingly”*

Authors' response: Thank you for pointing out this error; it has been corrected to “Notably” (line 134).

References

1. Sun Y, *et al.* B and T lymphocyte attenuator tempers early infection immunity. *Journal of immunology* **183**, 1946-1951 (2009).
2. Kobayashi Y, *et al.* B and T lymphocyte attenuator inhibits LPS-induced endotoxic shock by suppressing Toll-like receptor 4 signaling in innate immune cells. *Proceedings of the National Academy of Sciences of the United States of America* **110**, 5121-5126 (2013).
3. Cheng T, Bai J, Chung CS, Chen Y, Biron BM, Ayala A. Enhanced Innate Inflammation Induced by Anti-BTLA Antibody in Dual Insult Model of Hemorrhagic Shock/Sepsis. *Shock* **45**, 40-49 (2016).
4. Wu T, *et al.* Development of diagnostic criteria and a prognostic score for hepatitis B virus-related acute-on-chronic liver failure. *Gut* **67**, 2181-2191 (2018).
5. Xiang X, *et al.* Interleukin-22 ameliorates acute-on-chronic liver failure by reprogramming impaired regeneration pathways in mice. *Journal of hepatology* **72**, 736-745 (2020).
6. Sarin SK, *et al.* Acute-on-chronic liver failure: consensus recommendations of the Asian Pacific Association for the Study of the Liver (APASL) 2014. *Hepatology international* **8**, 453-471 (2014).
7. Sarin SK, *et al.* Acute-on-chronic liver failure: consensus recommendations of the Asian Pacific association for the study of the liver (APASL): an update. *Hepatology international* **13**, 353-390 (2019).
8. Moreau R, *et al.* Acute-on-chronic liver failure is a distinct syndrome that develops in patients with acute decompensation of cirrhosis. *Gastroenterology* **144**, 1426-1437, 1437.e1421-1429 (2013).
9. Bajaj JS, *et al.* Survival in infection-related acute-on-chronic liver failure is defined by extrahepatic organ failures. *Hepatology* **60**, 250-256 (2014).

REVIEWERS' COMMENTS

Reviewer #1 (expertise in immune checkpoint blockade and liver Immunology):

I would like to thank the authors for their efforts to address the points raised previously. I have no additional concerns.

Reviewer #2 (expertise in liver Immunology):

The authors have appropriately responded to the questions raised, and hereby significantly strengthened the findings and the manuscript.

Reviewer #3 (expertise in hepatology and liver failure):

In their paper Yu et al investigated the relevance of the B- and T-lymphocyte attenuator (BTLA) in T cell immune response with regard to infections and mortality risk in patients with HBV-related ACLF. As main findings of the original manuscript they showed that BTLA is significantly upregulated in peripheral and intrahepatic CD4+ T cells of patients with HBV-related ACLF and upregulation of BTLA expression correlated with disease progression, prognosis, and infectious complications in HBV-ACLF patients. Further characterization of BTLA+CD4+ T cells revealed that BTLA inhibits T cell activation and cytokine production but promotes CD4+ T cell apoptosis and exhaustion. Moreover, in an in vivo ACLF model immune exhaustion was seen in BTLA ko but not in WT mice.

The authors presented a methodologically sound manuscript with interesting mechanistic insights in the regulation and function of BTLA expression on CD4 T cells in HBV-ACLF.

However, several major aspects indicated necessity of major revision of the manuscript, especially regarding the ACLF classification based on the COSSH-ACLF scoring system which is different to EF CLIF criteria and the generality of the results of liver diseases other than chronic HBV-infection. In the revised version of their manuscript the authors address the points raised in my revision as follows:

1.1) As discussed previously, the COSSH score includes patients with HBV-related ACLF with and without liver cirrhosis meaning that the patients included in the presented study not necessarily have evidence of cirrhosis and therefore not necessarily fulfill EF CLIF ACLF criteria. In their rebuttal letter, the authors now have included another patient cohort (n=78) with patients who fulfill the EF CLIF ACLF diagnostic criteria. Within this cohort the authors still can show a significantly increased expression of BTLA on peripheral CD4+ T cells and can furthermore correlate the BTLA expression with disease severity and prognosis. By complementing this cohort, the authors satisfactorily confirm the relevance of BTLA also in patients who have been diagnosed with ACLF based on the EF CLIF criteria.

1.2) Regarding the generality of the results, the authors now have included a cohort of 14 patients with alcohol-induced cirrhosis and ACLF and could reproduce their findings. With this, they further strengthen the significance of the revised manuscript as the relevance of BTLA is not limited to patients HBV-induced chronic liver disease.

2. In the original manuscript the expression of BTLA on T-cells was only analyzed in patients with HBV-ACLF. The revised manuscript now includes the analysis of BTLA expression on T-cells in 14 patients with alcoholic cirrhosis or PBC and ACLF. For patients with alcoholic-induced ACLF a significant increase in the BTLA4 expression on T-cells could be confirmed, which is appreciated.

3. The revised version of the manuscript now provides the MFI data for HVEM expression, a BTLA-specific receptor, on CD80/86+ DCs, which is appreciated and further strengthens the relevance of BTLA in HBV-ACLF.

4. In the original manuscript the expression of CD25 was upregulated on CD4+ T cells in BTLA ko(!) mice which was confusing for the reader as the CD25 expression was presented to be a consequence of BTLA activation(!). The authors now present in vitro experiments where they applied functional antibodies that crosslink or activate BTLA signaling. The results showed that the crosslinked CD4+BTLA+ T cells expressed lower levels of CD25 than the non-crosslinked

CD4+BTLA+ T cells, which fits to the general assumption that the activation of BTLA results in increased expression of BTLA. The authors moreover confirm here that the differences in CD25 expression is mainly attributed to the difference between in vitro and in vivo experiments which is comprehensibly explained by the additional experiments.

The minor aspects including time point of PBMC sampling and group labeling have been satisfactorily addressed by the authors and do not need to undergo further revision/improvement.